# PerSense: Personalized Instance Segmentation in Dense Images

## Abstract

Leveraging large-scale pre-training, vision foundational models showcase notable performance benefits. Recent segmentation algorithms for natural scenes have advanced significantly. However, existing models still struggle to automatically segment personalized instances in dense and crowded scenarios, where severe occlusions, scale variations, and background clutter pose a challenge to accurately delineate densely packed instances of the target object. To address this, we propose **PerSense**, an end-to-end, training-free, and model-agnostic one-shot framework for **Per**sonalized instance **S**egmentation in d**ense** images. PerSense introduces a novel Instance Detection Module (IDM) that leverages density maps to encapsulate the spatial distribution of objects and automatically generate instance-level point prompts. To reduce false positives in these prompts, we design the Point Prompt Selection Module (PPSM), which refines the output of IDM. Both IDM and PPSM transforms density maps into precise point prompts, seamlessly integrate into our model-agnostic framework. Furthermore, we introduce a feedback mechanism which enables PerSense to improve the accuracy of density maps by automating the exemplar selection process for density map generation. Finally, To promote algorithmic advances and effective tools for this relatively underexplored task, we introduce PerSense-D, a diverse dataset exclusive to personalized instance segmentation in dense images. Our extensive experiments establish PerSense superiority in dense scenarios by achieving an mIoU of **71.61%** on PerSense-D, outperforming recent SOTA models by significant margins of **+47.16%**, **+42.27%**, **+8.83%**, and **+5.69%**. Additionally, our qualitative findings demonstrate the adaptability of our framework to images captured in-the-wild.

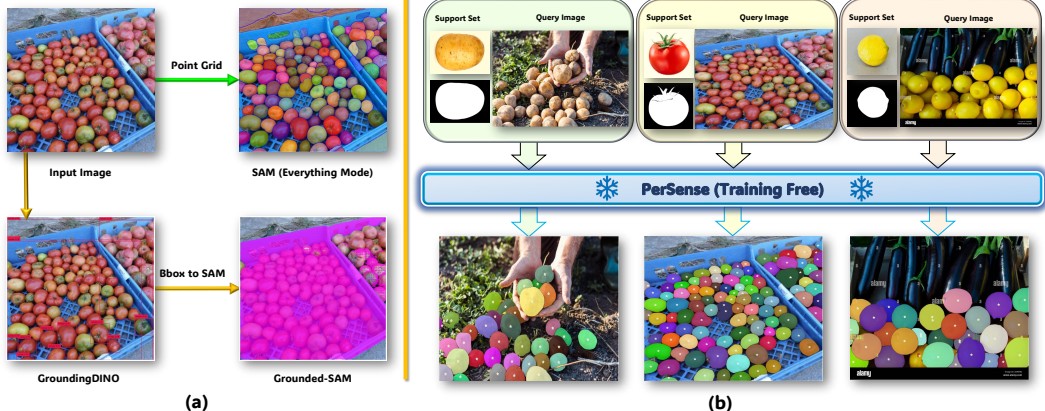

Figure 1: (a) Depicts the deteriorated segmentation performance of Grounded-SAM in dense scenario due to limitations associated with bounding box-based detections. Additionally, it demonstrates how SAM's "everything mode" indiscriminately segments both background and foreground, lacking any personalization for specific objects. (b) Introducing PerSense, a training-free and model-agnostic one-shot framework offering an end-to-end automated pipeline for personalized instance segmentation in dense images.

# 1 INTRODUCTION

Imagine working in a food processing sector where the goal is to automate the quality control process for vegetables, such as potatoes, using vision sensors. The challenge is to segment all potato instances in densely packed environments, where variations in scale, occlusions, and background clutter add complexity to the task. We refer to this task as *personalized instance segmentation in dense images* (Figure 1b), building on the concept of personalized segmentation, first introduced in Zhang et al. (2024). The term *personalized* refers to the segmentation of specific visual category within an image. Our task setting focuses on personalized instance segmentation, particularly in *dense scenarios*. To tackle this problem, a natural approach would be to explore the state-of-the-art (SOTA) segmentation models. One of the notable contributions in this domain is the Segment Anything Model (SAM) trained on the SA-1B dataset that consists of more than 1B masks derived from 11M images (Kirillov et al., 2023). SAM introduces a groundbreaking segmentation framework capable of generating masks for various objects in images using custom prompts, allowing for flexible segmentation across different visual elements. However, SAM lacks the inherent ability to segment distinct visual concepts as highlighted in Zhang et al. (2024). It primarily generates masks for individual objects using its "everything mode", which prompts the model with a point grid to segment all objects in the image, including both background and foreground (Figure 1a). Alternatively, users can manually draw a box or a point prompt to isolate specific instances. This process is labor-intensive, time-consuming, and hence not scalable for large-scale or automated applications.

One approach to achieve automation is to utilize the box prompts generated by a pre-trained object detector to isolate the object of interest. A recent work proposing an automated image segmentation pipeline is Grounded-SAM (Ren et al., 2024), which is a combination of open-vocabulary object detector GroundingDINO (Liu et al., 2023) and SAM (Kirillov et al., 2023). The underlying idea is to forward annotated bounding boxes from GroundingDINO to SAM for generating segmentation masks. However, bounding boxes are limited by box shape (fixed size anchors), occlusions (limited feature resolution), and the orientation of objects (Zand et al., 2021). In simpler terms, a standard bounding box (non-oriented and non-rotated) for a particular object may include portions of other instances. Additionally, when using non-max suppression (NMS), bounding box-based detections may group multiple instances of the same object together (Hosang et al., 2017), making it difficult to achieve proper delineation of object instances (Figure 1a). Although techniques like bipartite matching introduced in DETR (Carion et al., 2020) address the NMS issue but still bounding box-based detections are challenged due to variations in object scale, occlusions, and background clutter. These limitations become more pronounced when dealing with dense images (Wan & Chan, 2019).

Point-based prompting, mostly based on manual user input, is generally better than bounding box-based prompting for tasks that require high accuracy, fine-grained control, and the ability to handle occlusions, clutter, and dense instances (Maninis et al., 2018). However, the automated generation of point prompts using one-shot data, for personalized segmentation in dense scenarios, has largely remained unexplored. This motivates a novel segmentation framework specifically for dense images that can provide an automated pipeline capable of achieving instance-level segmentation through the generation of precise point prompts using one-shot data. Such capability will be pivotal for industrial automation, which uses vision-based sensors for applications such as object counting, quality control, and cargo monitoring. Beyond industrial automation, it could be transformative in the medical realm, particularly in tasks demanding segmentation at cellular levels. In such scenarios, relying solely on bounding box-based detections could prove limiting towards achieving desired segmentation accuracy.

We therefore approach this problem by exploring density estimation methods, which emphasize the spatial distribution of objects through the use of density maps (DM). While DMs are effective for calculating global object counts, they often fall short in providing precise point prompts for localization of individual objects at the instance level (Idrees et al., 2013). Although some studies have attempted to leverage DMs for instance segmentation in natural scenes (Cholakkal et al., 2019; Ma et al., 2015), there remains a potential gap for a streamlined approach that explicitly and effectively utilizes DM to achieve automated personalized instance segmentation in dense images. To this end, *our work introduces an end-to-end, training-free, and model-agnostic one-shot framework titled PerSense* (Figure 2). First, we develop a new baseline capable of automatically generating instance-level point prompts. This new baseline features a proposed Instance Detection Module (IDM) which leverages DMs to provide candidate point prompts. We generate DMs using a density map genera-

tor (DMG) which highlights spatial distribution of object of interest based on input exemplars. To allow automatic selection of effective exemplars for DMG, we automate the mostly manual process via a class-label extractor (CLE) and a grounding detector. Second, we design a Point Prompt Selection Module (PPSM) to mitigate false positives within the candidate point prompts. The proposed IDM and PPSM are essentially plug-and-play components and seamlessly integrate with our model-agnostic PerSense framework. Lastly, we introduce a robust feedback mechanism, which automatically refines the initial exemplar selection by identifying multiple rich exemplars for DMG based on the initial segmentation output of PerSense.

Finally, to our knowledge, there exists no dataset specifically targeting segmentation in dense images. While some images in mainstream segmentation datasets like COCO (Lin et al., 2014), LVIS (Gupta et al., 2019), and FSS-1000 (Li et al., 2020), may contain multiple instances of the same object category, the majority do not qualify as dense images due to the limited number of object instances. For example, images in the LVIS dataset contain an average of 11.2 instances across 3.4 object categories, resulting in about 3.3 instances per category. This low instance count is insufficient to represent dense scenarios in images. Therefore we introduce PerSense-D, a personalized one-shot segmentation dataset exclusive to dense images. PerSense-D comprises 717 dense images distributed across 28 diverse object categories with an *average count of 39 object instances per image*. These images present significant occlusion and background clutter, making our dataset a unique and challenging benchmark for enabling algorithmic advances and practical tools targeting personalized segmentation in dense images.

We report results on this newly introduced PerSense-D dataset, comparing PerSense with several SOTA segmentation models, including PerSAM (Zhang et al., 2024), Matcher (Liu et al., 2024), and Grounded-SAM (Ren et al., 2024). Our extensive experiments demonstrate PerSense's superior performance and efficiency in dense scenarios.

## 2 RELATED WORK

**One-shot personalized segmentation:** As discussed in sec 1, SAM (Kirillov et al., 2023) segmentations lack semantic meaning, which limits it in segmenting personalized visual concepts. To overcome this challenge, PerSAM is introduced in Zhang et al. (2024), which offers a training-free automated framework for one-shot personalized segmentation using SAM. PerSAM performs well in segmenting few instances of similar category, efficiently distinguishing and segmenting objects through its iterative masking approach. However, when applying PerSAM to dense images with many instances of the same object, several challenges may arise. Firstly, its iterative masking strategy, which segments objects one by one, can become computationally expensive and slow, as the number of iterations is proportional to the number of object instances in the image. Moreover, the confidence map's accuracy may degrade as more objects are masked out, making it difficult to distinguish between closely packed or overlapping instances. Also, the confidence thresholding strategy introduced in PerSAM, which halts the process when the confidence score drops below a set threshold, may lead to premature termination of segmentation process, even when valid objects are still present (see sec 5). Unlike PerSAM, our PerSense utilizes DM to generate precise instance-level point prompts in a single iteration.

Matcher introduced in Liu et al. (2024) integrates a versatile feature extraction model with a class-agnostic segmentation model and leverages bidirectional matching to align semantic information across images for tasks like semantic segmentation and dense matching. However, its instance-level matching capability inherited from the image encoder is relatively limited, which hampers its performance for instance segmentation tasks. Matcher employs reverse matching to eliminate outliers and uses K-means clustering for instance-level sampling, which can become a bottleneck in dense and cluttered scenes due to challenges posed by varying object scales. Additionally, Matcher forwards the bounding box of the matched region as a box prompt to SAM, which can have adverse affect due to the limitations of box-based detections, especially in crowded environments. To address these challenges, PerSense utlizes DMG to obtain a personalized DM which obviates the need for clustering and sampling. With IDM and PPSM, PerSense accurately generates at least one point prompt for each detected instance.

Another one-shot segmentation method, SLiMe (Khani et al., 2023), enables personalized segmentation based on segmentation granularity in the support set, rather than object category. Despite

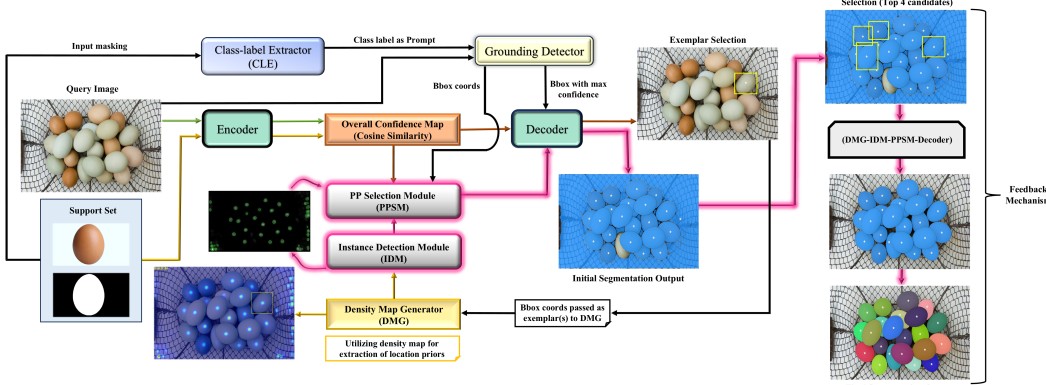

Figure 2: **Overall architecture:** PerSense is a one-shot framework for personalized instance segmentation in dense images. It begins by extracting class-label using CLE followed by exemplar selection to generate density maps using DMG. IDM identifies candidate point prompts from these density maps, which are subsequently refined using the PPSM. The feedback mechanism identifies high-quality exemplars using the initial segmentation output from the decoder and leverages them to refine initial density maps. The point prompts generated from these refined maps enables PerSense to achieve precise personalized instance segmentation in dense and cluttered scenes.

its strong performance, SLiMe tends to produce noisy segmentations for small objects due to the smaller attention maps extracted from Stable Diffusion (Rombach et al., 2022) compared to the input image. Given our focus on instance segmentation in dense images with varying object scales, SLiMe may not be the most suitable choice.

**Interactive segmentation:** Recently, the task of interactive segmentation has received a fine share of attention. Works like InterFormer (Huang et al., 2023), MIS (Li et al., 2023) and SEEM (Zou et al., 2024) provide a user-friendly interface to segment an image at any desired granularity, however, these models are not scalable as they are driven by manual input from the user.

## 3 METHOD

We introduce PerSense, a training-free and model-agnostic one-shot framework designed for personalized instance segmentation in dense images (Figure 2). Here, we describe the core components of our PerSense framework, including class-label extraction using CLE and exemplar selection for DMG (sec. 3.1), IDM (sec. 3.2), PPSM (sec. 3.3), and the feedback mechanism (sec. 3.4). See Appendix A.1 for the overall pseudo-code of PerSense.

### 3.1 CLASS-LABEL EXTRACTION AND EXEMPLAR SELECTION FOR DMG

PerSense operates as a one-shot framework, wherein a support set is utilized to guide the personalized segmentation of an object in the query image that shares semantic similarity with the support object. Initially, input masking is applied to the support image using the coarse support mask to isolate the object of interest. The resulting input masked image is fed into the CLE with a custom prompt, "*Name the object in the image?*". The CLE generates a description of the object in the image, from which the noun is extracted, representing the object category. Subsequently, the grounding detector is prompted with this class-label to facilitate personalized object detection in the query image. To enhance the prompt, we prefixed the term "all" with the class-label.

Next, we compute the cosine similarity score $S_{score}$ between query $Q$ and support $S_{supp}$ features coming from the encoder as follows:

$$S_{score}(Q, S_{supp}) = \text{cos\_sim}(f(Q), f(S_{support})), \quad (1)$$

where $f(\cdot)$ represents the encoder. Utilizing this score along with detections from the grounding object detector, we extract the positive location prior. Specifically, we identify the bounding box

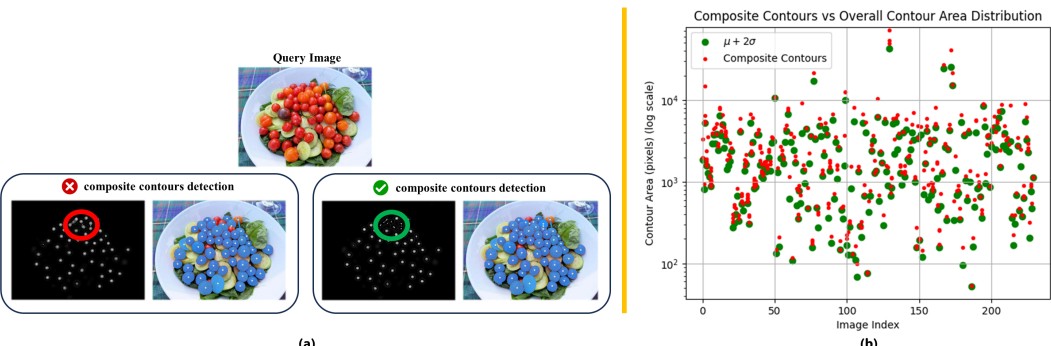

(a)                                                    (b)

Figure 3: (a) Without identifying composite contours, multiple object instances may be incorrectly grouped (red circle). Identification of composite contours (green circle) enables accurate localization of child contours (missed detections). (b) The plot illustrates the presence of composite contours beyond $\mu + 2\sigma$ in the contour area distribution for 250 images in the PerSense-D dataset.

$B_{max}$ with the highest detection confidence and proceed to locate the pixel-precise point $P_{max}$ with the maximum similarity score within this bounding box:

$$P_{max} = \arg \max_{P \in B_{max}} S_{score}(P, S_{supp}),\qquad(2)$$

where $P$ represents candidate points within the bounding box $B_{max}$. This identified point serves as the positive location prior, which is subsequently fed to the decoder for segmentation. Additionally, we extract the bounding box surrounding the segmentation mask of the object. This process effectively refines the original bounding box provided by the grounding detector. The refined bounding box is then forwarded as an exemplar to the DMG for generation of density map.

## 3.2 INSTANCE DETECTION MODULE (IDM)

The IDM begins by converting the DM from the DMG into a grayscale image $I_{gray}$. Next, a binary image $I_{binary}$ is created from $I_{gray}$ using a pixel-level threshold $T$ ($T \in [0, 255]$):

$$I_{binary}(x, y) = \begin{cases} 1 & \text{if } I_{gray}(x, y) \geq T \\ 0 & \text{if } I_{gray}(x, y) < T \end{cases}\qquad(3)$$

for all pixels $(x, y)$ in the image, where $I_{binary}$ is the resulting binary image. A morphological erosion operation is then applied to $I_{binary}$ using a $3 \times 3$ kernel $K$:

$$I_{eroded}(x, y) = \min_{(i,j) \in K} I_{binary}(x + i, y + j),\qquad(4)$$

where $I_{eroded}$ is the eroded image, and $(i, j)$ iterates over the kernel $K$ to refine the boundaries and eliminate noise from the binary image. We deliberately used a small kernel to avoid damaging the original densities of true positives. Next, contours are identified in the eroded binary image, and for each contour $C$, its area $A_C$ and center pixel coordinates $(x_C, y_C)$ are computed. We calculate the mean $\mu$ and standard deviation $\sigma$ of all contour areas to assess the distribution of contour sizes:

$$\mu = \frac{1}{N} \sum_{i=1}^{N} A_{C_i}, \quad \sigma = \sqrt{\frac{1}{N} \sum_{i=1}^{N} (A_{C_i} - \mu)^2},\qquad(5)$$

where $N$ is the total number of contours. Subsequently, composite contours, which represent multiple objects in one contour, are detected using a threshold based on the distribution of contour sizes. This is necessary to identify the regions that are detected as one contour but encapsulate multiple instances of the object of interest (Figure 3a). Such regions are scarce and can be detected as outliers, essentially falling beyond $\mu + 2\sigma$, considering the contour size distribution (Figure 3b). For each detected composite contour, a distance transform is applied to expose child contours for ease of detection. Finally, the algorithm returns the center points obtained from all detected contours (parent and child) as candidate point prompts. In summary, through systematic analysis of the DM, IDM identifies regions of interest and generates candidate point prompts, which are subsequently forwarded to PPSM for final selection. See Appendix A.1 for pseudo-code of IDM.

### 3.3 POINT PROMPT SELECTION MODULE (PPSM)

The PPSM serves as a critical component in the PerSense pipeline, tasked with filtering candidate point prompts for final selection. For each candidate point prompt received from IDM, we compare the corresponding query-support similarity score using an adaptive threshold defined as:

$$\text{sim\_threshold} = \frac{\text{max\_score}}{\text{object\_count}/\text{norm\_const}} \quad (6)$$

where *max_score* is the maximum value of the query-support similarity score, the *object_count* corresponds to the number of instances of the desired object present in the query image, and the *norm_const* is a normalization factor, set as $\sqrt{2}$ to make the threshold adaptive with respect to the object count (see sec 5.1). A fixed similarity threshold would struggle in this case, as the query-support similarity score varies significantly even with small intra-class variations. Moreover, for highly crowded images (*object_count* $> 50$), the similarity score for positive location priors can vary widely, necessitating an adaptive threshold that accounts for the density (count) of the query image. In other words, as object instances increase, the query-support similarity score distribution widens due to intra-class variations. To address this challenge, our adaptive threshold is based on the maximum query-support similarity score as well as the object count within the query image. In addition to this, PPSM leverages bounding box data from the grounding detector to ensure filtered point prompts fall within the box boundaries. These filtered points are then passed to the decoder for segmentation. See Appendix A.1 for pseudo-code of PPSM.

### 3.4 FEEDBACK MECHANISM

PerSense proposes a feedback mechanism to enhance the exemplar selection process for the DMG by leveraging the initial segmentation output from the decoder. Let $M_{seg}$ represent the initial segmentation mask generated by the decoder, and let $S_{mask}$ denote the mask scores provided by SAM.

$$C_{Top} = \text{Top\_k}(M_{seg}, S_{mask}, k), \quad (7)$$

where $C_{Top}$ represents the set of the top $k$ candidates selected based on their mask scores. In our case $k = 4$ (see sec 5.1). These selected candidates are then forwarded as exemplars to DMG in a feedback manner. This leads to improved accuracy of the DM and consequently enhances the segmentation performance. The quantitative analysis of this aspect is further discussed in sec 5, which explicitly highlights the value added by the proposed feedback mechanism.

## 4 NEW DATASET (PERSENSE-D)

PerSense utilizes DMs generated by DMG for point prompt extraction via IDM and PPSM, specifically for dense images containing several instances of the same object. While existing segmentation datasets like COCO (Lin et al., 2014), LVIS (Gupta et al., 2019), and FSS-1000 (Li et al., 2020) may contain some images with multiple instances of the same object category, the majority of images do not represent dense scenarios due to limited or few object instances. For example, on average each image in LVIS (Gupta et al., 2019) is annotated with 11.2 instances from 3.4 object categories. This results in an average of 3.3 instances per single category, which is insufficient to represent dense scenarios in images. To address this, we introduce PerSense-D, a diverse dataset exclusive to segmentation in dense images. PerSense-D comprises 717 images distributed across 28 object categories, with an average count of 39 objects per image. The dataset is designed to serve as a challenging benchmark for driving algorithmic innovations while facilitating the development of practical tools across diverse domains such as medical, agriculture, environmental monitoring, and autonomous systems. Given our focus on one-shot personalized dense image segmentation, we explicitly supply 28 support images labeled as "00", each containing a single object instance intended for personalized segmentation in the corresponding object category. This can facilitate fair evaluation among various one-shot approaches as no random seeding is required.

**Image Collection and Retrieval:** Out of 717 images, we have 689 dense query images and 28 support images. To acquire the set of 689 dense images, we initiated the process with a collection of candidate images obtained through keyword searches. To mitigate bias, we retrieved the candidate images by querying object keywords across three distinct Internet search engines: Google, Bing,

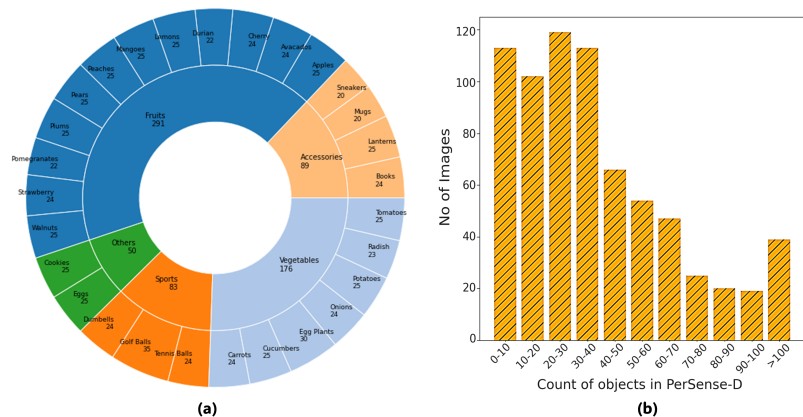

Figure 4: (a) Object categories in PerSense-D. (b) No of images vs range bins of object count.

and Yahoo. To diversify the search query keywords, we prefixed adjectives such as '*multiple*', '*lots of*', and '*many*' before the category names. In every search, we collected the first 100 images that fall under CC BY-NC 4.0 licensing terms. With 28 categories, we gathered a total of 2800 images, which were subsequently filtered in the next step.

**Manual Inspection and Filtering:** The candidate images were manually inspected following a three-point criterion. (1) The image quality and resolution should be sufficiently high to enable easy differentiation between objects. (2) Following the criterion in object counting dataset FSC-147 (Ranjan et al., 2021), we set the minimum object count to 7 per image for our PerSense-D benchmark. (3) The image shall contain a challenging dense environment with sufficient occlusions among object instances along with background clutter. Based on this criterion, we filtered 689 images out of 2800 candidates.

**Semi-automatic Image Annotation Pipeline:** We crowdsourced the annotation task under appropriate institutional approval. We devised a semi-automatic annotation pipeline. Following the *model-in-the-loop* strategy outlined in Kirillov et al. (2023), we utilized our PerSense to provide an initial segmentation mask. This initial mask was then manually refined and corrected by annotators using pixel-precise tools such as the OpenCV image annotation tool and Photoshop's "quick selection" and "lasso" tool, which allows users to loosely select an object automatically. As the images were dense, the average time to manually refine single image annotation was around 15 minutes.

**Dataset Statistics:** The dataset contains a total of 717 images (689 query and 28 support images). Average count is 39 objects per image, with a total of 28,395 objects across the entire dataset. The minimum and maximum number of objects in a single image are 7 and 218, respectively. The average resolution (h × w) of images is 839 × 967 pixels. Figure 4 presents detail of object categories in PerSense-D and a histogram depicting the number of images across various ranges of object count.

## 5 EXPERIMENTS

**Implementation Details and Evaluation Metrics:** Our PerSense is model-agnostic and leverages a CLE, grounding detector, and DMG for personalized instance segmentation in dense images. For CLE, we leverage VLM as it is best suited for this task. We follow VIP-LLaVA (Cai et al., 2024), which utilizes CLIP-336px (Radford et al., 2021) and Vicuna v1.5 (Chiang et al., 2023) as visual and language encoders, respectively. We use GroundingDINO (Liu et al., 2023) as the grounding detector. To demonstrate model-agnostic capability of PerSense, we separately utilize DSALVANet (He et al., 2024) and CounTR (Liu et al., 2022) pretrained on FSC-147 dataset (Ranjan et al., 2021) as DMG. Finally, we utilize SAM (Kirillov et al., 2023) encoder and decoder for personalized segmentation following the approach in (Zhang et al., 2024). We evaluate segmentation performance on the PerSense-D dataset specifically created for dense scenarios. We use standard evaluation metric of mIoU (mean Intersection over Union) for evaluating segmentation performance. **No training is involved in any of our experiments.**

Table 1: We compare overall mIoU between PerSense and SOTA methods on PerSense-D dataset. ‡ indicates training-free methods. * denotes that PerSAM's inference time is calculated as (number of object instances × 1.02) sec. Given that the PerSense-D dataset contains an average of 39 object instances per image, the average inference time for PerSAM is (39 × 1.02) = 39.78 sec. † indicates that PerSAM-F requires an average of 8 seconds of training time per class, which is added to the training-free inference time and incurred once per class.

| Method | Venue | mIoU | Avg inference time (per image) (sec) |
|---|---|---|---|
| PerSAM‡ (Zhang et al., 2024) | ICLR'24 | 24.45 | 39.78* |
| PerSAM-F (Zhang et al., 2024) | ICLR'24 | 29.34 | $(39.78 + 8)^{\dagger}$ |
| Matcher‡ (Liu et al., 2024) | ICLR'24 | 62.78 | 10.2 |
| Grounded-SAM‡ (Ren et al., 2024) | arXiv'24 | 65.92 | 1.8 |
| **PerSense‡ (DMG: DSALVANet)** | this work | **70.96** | 2.7 |
| **PerSense‡ (DMG: CounTR)** | this work | **71.61** | |

**Results:** We compare our PerSense with a variety of generalist models like PerSAM (Zhang et al., 2024), Matcher (Liu et al., 2024) and Grounded-SAM (Ren et al., 2024) utilizing PerSense-D as evaluation benchmark. To be fair in comparison with Grounded-SAM, we ensured that all classes in PerSense-D overlaps with at least one of the datasets on which GroundingDINO is pre-trained. Importantly, all the classes in PerSense-D are common in Objects365 dataset (Shao et al., 2019). The class-label extracted by CLE in PerSense (sec 3.1), using one-shot data, is also fed to Grounded-SAM for personalized segmentation. We report the results in Table 1. Our PerSense achieves **71.61%** mIoU, surpassing PerSAM, PerSAM-F, Matcher and Grounded-SAM by a significant margin of **+47.16%**, **+42.27%**, **+8.83%** and **+5.69%**, respectively. Figure 6 showcases our qualitative results. For qualitative analysis of PerSense at each step, please see Appendix A.2.

**Discussion:** We observed that the decline in PerSAM's segmentation performance on PerSense-D is mainly due to the premature termination of the segmentation process (see Figure 6), influenced by its naive confidence thresholding strategy. In dense environments with closely packed instances of similar objects, the confidence scores can drop below the fixed set threshold, particularly as the confidence map becomes noisy and less clear after multiple masking iterations. This leads the algorithm to misidentify remaining instances as background, resulting in premature termination of the segmentation process and ultimately compromising the accuracy of the segmentation outcomes.

For Matcher, we observed that in dense scenarios, the patch-level matching and correspondence matrix struggles to identify distinct regions when there is significant overlap or occlusion among objects. Additionally, Matcher uses a bidirectional matching strategy that, while effective in less crowded scenes, can introduce false positives in densely packed environments, where minor differences in appearance between objects are hard to capture.

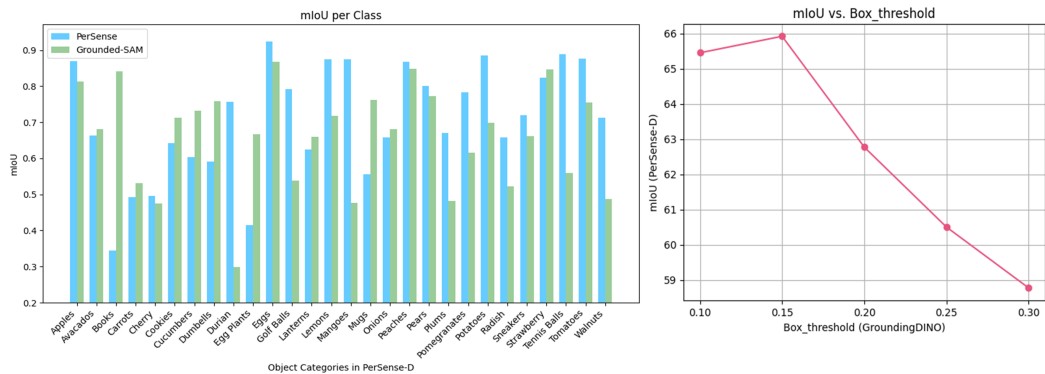

Figure 5: Left: Class-wise mIoU comparison between PerSense and Grounded-SAM on PerSense-D. Right: Grounded-SAM mIoU on PerSense-D vs Box_threshold in GroundingDINO

We present a class-wise comparison of mIoU on PerSense-D considering PerSense and Grounded-SAM (Figure 5, Left). PerSense excels in accurately segmenting object categories like "Durian," "Mangoes," and "Walnuts," where there is minimal demarcation between instances. In contrast, Grounded-SAM often missegments undesired regions between instances due to its reliance on bounding box-based detections. However, for categories with zero separation between instances or tightly merged flat boundaries, such as "Books," Grounded-SAM performs better, as distinguishing distinct instances without clear boundaries is challenging for PerSense. Additionally, for categories with significant intra-class variation, like "Eggplants," "Cookies," "Cucumbers," and "Dumbbells," PerSense shows a relative decline in performance compared to Grounded-SAM, as its one-shot context provides access to limited object features.

We evaluated the runtime efficiency of all methods using the PerSense-D dataset on a single NVIDIA GeForce RTX 4090 GPU with a batch size of 1. As reported in Table 1, PerSAM is computationally inefficient due to its iterative masking strategy, which requires as many iterations as there are object instances in the image. Matcher averages 10.2 seconds per image, which limits its suitability for applications demanding fast inference. PerSense, by contrast, takes an average of 2.7 seconds per image, while Grounded-SAM takes about 1.8 seconds under similar conditions. This relative temporal overhead of 0.9 seconds for PerSense is mainly attributed to the generation of DMs for extracting instance-level point prompts in dense scenarios. In summary, PerSense introduces minimal latency compared to Grounded-SAM at the cost of improved segmentation performance, while being significantly more efficient and accurate than the recent SOTA methods, PerSAM and Matcher.

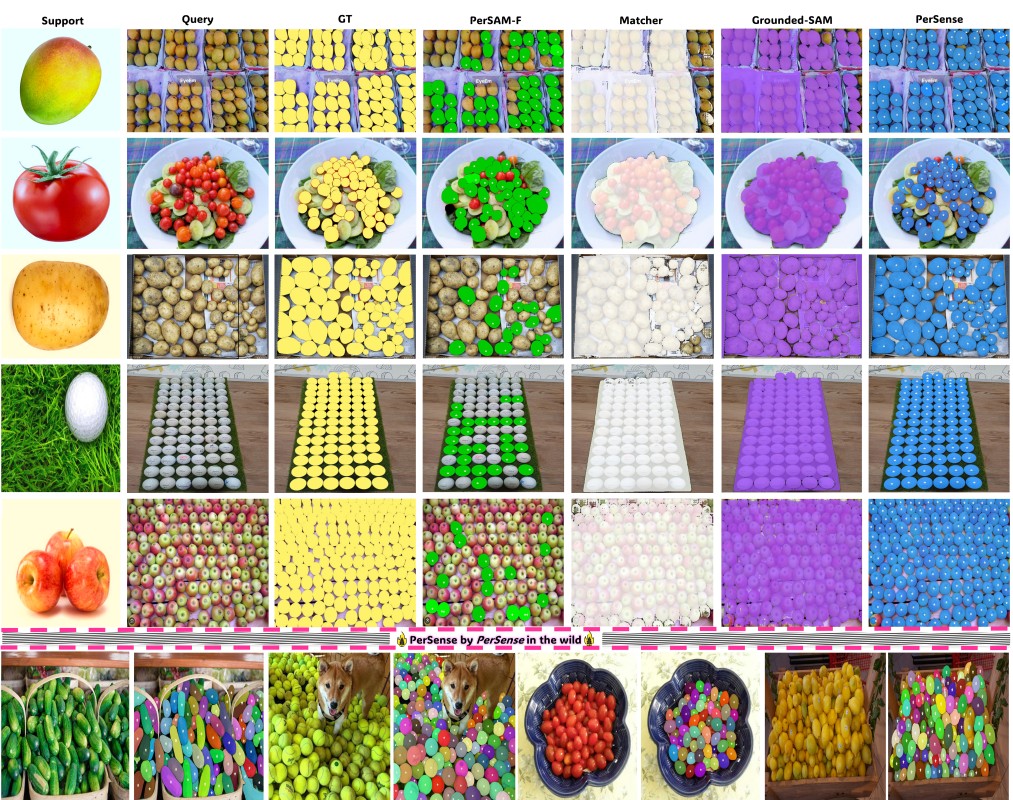

Figure 6: Qualitative comparison of PerSense with SOTA

## 5.1 ABLATION STUDY

**Component-wise Ablation Study of PerSense:** The proposed PerSense framework includes three key components: IDM, PPSM, and a feedback mechanism. An ablation study was conducted to highlight PerSense's model-agnostic capability and assess each component's contribution to performance using two different DMGs, DSALVANet and CounTR (Table 2a). Even with the baseline network, PerSense (CounTR) outperformed Grounded-SAM by **+2.2%**, while PerSense (DSALVANet)

Table 2: (a) Component-wise ablation study of PerSense. (b) Choice of normalization factor for adaptive threshold in PPSM. (c) Varying shots of exemplar data in DMG using feedback mechanism.

| | (a) | | | (b) | | (c) | |
|---|---|---|---|---|---|---|---|
| **Modules** | **baseline** | **baseline + PPSM** | **PerSense** | **Norm Factor** | **mIoU** | **No. of Shots** | **mIoU** |
| IDM | yes | yes | yes | 1 | 70.41 | 1-shot | 65.78 |
| PPSM | no | yes | yes | $\sqrt{2}$ | **70.96** | 2-shot | 69.24 |
| Feedback | no | no | yes | $\sqrt{3}$ | 69.59 | 3-shot | 70.53 |
| DMG: DSALVANet | 65.58 | 66.95 | **70.96** | $\sqrt{5}$ | 68.95 | **4-shot** | **70.96** |
| mIoU(Gain) | (-) | (+1.37) | (+4.01) | | | 5-shot | 70.90 |
| DMG: CounTR | 68.12 | 70.58 | **71.61** | | | 6-shot | 70.81 |
| mIoU(Gain) | (-) | (+2.46) | (+1.03) | | | | |

showed comparable performance. Adding PPSM improved mIoU by **+1.37%** for DSALVANet and **+2.46%** for CounTR, with CounTR's higher increase indicating the presence of relatively more false positives in its DMs, despite better localization. This aligns with the findings in He et al. (2024), which report lower performance of CounTR relative to DSALVANet for few-shot object counting task. Finally, the feedback mechanism improved mIoU by **+4.01%** for DSALVANet and **+1.03%** for CounTR, indicating DSALVANet's sensitivity to exemplar selection for accurate DM generation.

**Varying the Detection Threshold in Grounding Detector:** We conducted an ablation study to assess the impact of varying the detection threshold in GroundingDINO on segmentation performance for Grounded-SAM (Figure 5, Right). The bounding box threshold was varied from 0.10 to 0.30 in increments of 0.05. For comparison with PerSense, we selected 0.15 as the optimal threshold, as it achieved the highest mIoU for Grounded-SAM on the PerSense-D benchmark. To ensure fairness, we applied the same threshold for GroundingDINO within the PerSense framework.

**Choice of Normalization Factor for Adaptive Threshold in PPSM:** For the adaptive threshold in PPSM, we tested different values of the normalization constant. Empirical results (Table 2b), demonstrate that $\sqrt{2}$ is the optimal choice, as it led to the most significant performance improvements in the overall mIoU evaluation.

**Varying No of Shots for Exemplar Data in Feedback Mechanism:** We automated the selection of the best exemplars for DMG based on SAM scores using the proposed feedback mechanism. As shown in Table 2c, segmentation performance on PerSense-D saturates after 4-shot, as additional exemplars do not provide any new significant information about the object of interest.

## 6 CONCLUSION

We presented PerSense, a training-free and model-agnostic one-shot framework for personalized instance segmentation in dense images. We proposed IDM and PPSM, which transforms density maps from DMG into personalized instance-level point prompts for segmentation. We also proposed a robust feedback mechanism in PerSense which automates and improves the exemplar selection process in DMG. Finally to promote algorithmic advancements considering the persense task, we presented PerSense-D, a dataset exclusive to personalized segmentation in dense images and established superiority of our method on this benchmark by comparing it with the SOTA.

**Limitations and Broader Impact:** PerSense is specifically designed for dense images, deriving point prompts from density maps generated by DMG. Therefore, it would not be fair to gauge PerSense performance on standard segmentation datasets with few object instances. In such cases, traditional object detection methods are more effective due to fewer occlusions and easier object boundary delineation, rendering density map generation inefficient. While PerSense employs IDM and PPSM to refine density maps and reject false positives, respectively, it cannot recover any true positives missed initially by DMG, during generation of DMs (see Appendix A.3). Being training-free and built upon open-source models, PerSense significantly reduces carbon emissions. Presently, no notable ethical or social implications are anticipated from our work.

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

# A APPENDIX

## A.1 ALGORITHMS

---

**Algorithm 1:** PerSense

---

**Input:** *Query Image ($I_Q$), Support Image ($I_S$), Support Mask ($M_S$)*
**Output:** *Segmentation Mask*

1 Perform input masking: $I_{\text{masked}} = I_S \odot M_S$;
2 Extract class-label using CLE from $I_{\text{masked}}$ (text prompt: "*Name the object in the image?*");
3 Prompt grounding detector with class-label;
4 Obtain grounded detections;
5 Bounding box with max confidence $\rightarrow$ decoder;
6 Obtain segmentation mask of the object;
7 Refine bounding box coordinates using the segmentation mask;
8 Exemplar Selection: Refined bounding box $\rightarrow$ DMG;
9 Obtain DM from DMG;
10 Process DM using IDM to generate candidate point prompts ($PP_{\text{cand}}$);
11 $PP_{\text{cand}} \rightarrow$ PPSM $\rightarrow$ final point prompts ($PP_{\text{final}}$);
12 $PP_{\text{final}} \rightarrow$ decoder;
13 Obtain an initial segmentation output;
14 Select Top 4 candidates as DMG exemplars based on SAM score;
15 Feedback: Repeat Steps 8 to 13;
16 Obtain final segmentation output;

---

---

**Algorithm 2:** Instance Detection Module (IDM)

---

**Input:** *Density Map (DM) from DMG*
**Output:** *Candidate Point Prompts (PP)*

1 Convert DM to grayscale image ($I_{gray}$);
2 Threshold to binary (threshold = 30) to obtain binary image ($I_{binary}$);
3 Erode $I_{binary}$ using $3 \times 3$ kernel;
4 Find BLOB_contours ($C_{BLOB}$) in the eroded image ($I_{eroded}$);
5 **for** *contour in $C_{BLOB}$* **do**
6     Compute contour area ($A_{contour}$);
7     Find center pixel coordinates for each contour;
8 **end**
9 Compute mean ($\mu$) and standard deviation ($\sigma$) using $A_{contour}$;
10 Detect composite_contours ($C_{composite}$) by thresholding $A_{contour}$;
11     area_threshold = $\mu + 2\sigma$;
12 **for** *contour in $C_{BLOB}$* **do**
13     Compute $A_{contour}$;
14     **if** *$A_{contour} >$ area_threshold* **then**
15        save contour as $C_{composite}$;
16     **end**
17 **end**
18 **for** *contour in $C_{composite}$* **do**
19     Apply distance transform [threshold = *0.5 * dist_transform.max()*];
20     Find child contours;
21     Find center pixel coordinates for each child contour;
22 **end**
23 **return** center points from Steps 7 and 21 as candidate PP;

---

---

**Algorithm 3:** Point Prompt Selection Module (PPSM)

---

**Input:** candidate_PP, similarity_matrix, object_count, grounded_detections

**Output:** selected_PP

1 max_score ← Get the maximum similarity score from similarity_matrix;

2 selected_PP ← [ ];                    // Empty list to store selected_PP

3 sim_threshold ← max_score / (object_count / $\sqrt{2}$);

4 **for** *each PP in candidate_PP* **do**

5      PP_similarity ← similarity_matrix(PP);

6      **for** *each box in grounded_detections* **do**

7          **if** *(PP_similarity > sim_threshold)* **and** *(PP lies within box)* **then**

8              selected_PP.append(PP);

9          **end**

10      **end**

11 **end**

12 **return** selected_PP;

---

## A.2 Step-wise Qualitative Analysis of PerSense

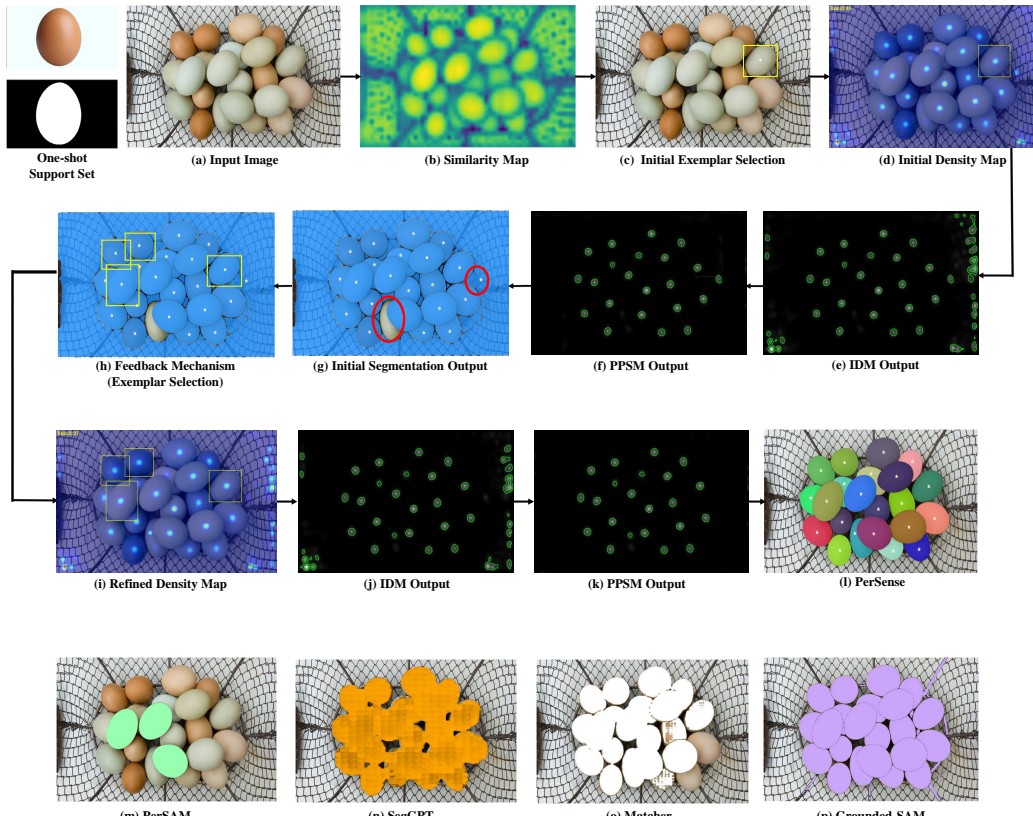

Figure 7: Step-by-step qualitative analysis of the PerSense framework. Starting with an input image (a), a cosine similarity map (b) is generated using the support set. Leveraging this similarity map and the output from the grounding detector, exemplar selection (c) is carried out to obtain an initial density map (d) utilizing the DMG. For the given example, the object class is "egg" and the ground truth object count is 22. As can be seen in (d), the initial density map estimates the object count as 45 with an error count of 23 ($45 - 22 = 23$). This initial density map is then processed by IDM to generate candidate point prompts (e), which are refined by PPSM to filter false positives, resulting in final point prompts (f). These point prompts are then forwarded to decoder to obtain an initial segmentation output (g). It can be observed in (g) that PPSM effectively eliminated the majority of false positives; however, a few still remain alongside false negatives (highlighted in red circle). This initial segmentation output is utilized by the feedback mechanism to refine exemplar selection based on SAM scores (h), resulting in a more accurate density map (i). As can be observed that the refined density map predicts object count as 37, reducing the error count from 23 to 15. Next, the IDM and PPSM modules subsequently leverage the refined density map to generate precise point prompts represented by (j) and (k), respectively. This enables PerSense to perform personalized instance segmentation in dense images given as (l). For comparison, outputs of PerSAM, SegGPT, Matcher, and Grounded-SAM are shown in (m), (n), (o), and (p), respectively.

## A.3 FAILURE CASES

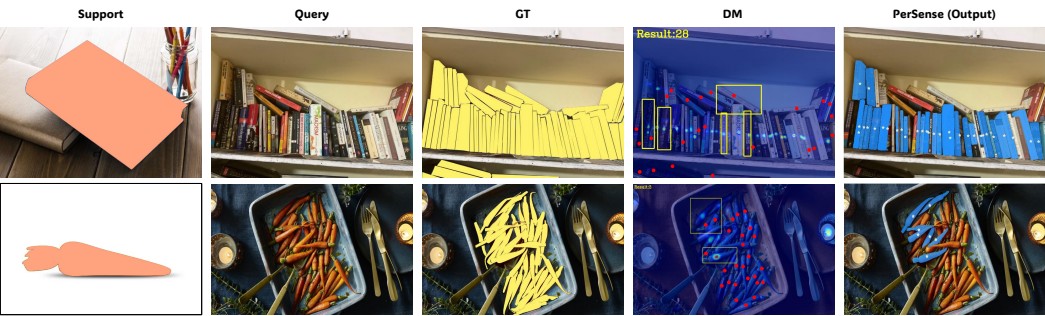

Figure 8: The figure illustrates scenarios where PerSense's performance deteriorates, primarily due to its reliance on the generated density map. In the first row, where the goal is to segment all instances of the "book" class, the density map excludes many true positives (highlighted in red), which PerSense cannot recover once they are lost during DM generation. A similar issue is seen in the second row, where a poor-quality density map for the "carrot" class leads to missed instances, negatively impacting PerSense segmentation performance.

## B MATHEMATICAL INSIGHTS INTO IDM AND PPSM

To enhance understanding of the design of our proposed modules, we provide additional mathematical insights into IDM and PPSM. Starting with IDM, we discuss the mathematical framework for composite contour detection using statistical thresholding, progressing to the computation of centroids for candidate point prompts. For PPSM, we offer a theoretical rationale behind the formulation of the adaptive threshold.

### B.1 INSTANCE DETECTION MODULE (IDM)

**Contour Detection and Area Calculation:** Contours are identified from the binary image, and the area $A_{\text{contour}}$ of each contour is calculated. Assuming that the contour areas follow a Gaussian (Normal) distribution, we define:

$$A_{\text{contour}} \sim \mathcal{N}(\mu, \sigma^2)$$

where:

- $\mu$ is the mean area of contours, representing typical object size.
- $\sigma$ is the standard deviation, representing variation in contour areas due to size differences among single instances.

The mean $\mu$ and standard deviation $\sigma$ are computed as:

$$\mu = \frac{1}{N} \sum_{i=1}^{N} A_{\text{contour}_i}, \quad \sigma = \sqrt{\frac{1}{N} \sum_{i=1}^{N} (A_{\text{contour}_i} - \mu)^2}$$

where $N$ is the number of detected contours.

**Composite Contour Detection using Statistical Thresholding:** To distinguish single-instance contours from composite contours, we set an adaptive threshold based on the Gaussian distribution properties:

$$T_{\text{composite}} = \mu + 2\sigma$$

This threshold was adopted based on statistical analysis presented in Figure 3(b) of the paper which illustrates the presence of composite contours beyond $\mu + 2\sigma$ in the contour area distribution for 250

images in the PerSense-D dataset. The composite threshold $T_{\text{composite}}$ captures unusually large contours, likely representing composite regions where multiple objects are clustered together. Contours with areas exceeding $T_{\text{composite}}$ are flagged as composite:

$$A_{\text{composite}} = \{A_{\text{contour}} \mid A_{\text{contour}} > T_{\text{composite}}\}$$

The probability of a contour being composite can be calculated as:

$$P(A_{\text{contour}} > T_{\text{composite}}) = 1 - \Phi\left(\frac{T_{\text{composite}} - \mu}{\sigma}\right)$$

where $\Phi$ is the cumulative distribution function of the standard normal distribution.

**Distance Transform for Child Contour Detection:** For each composite contour, we apply a distance transform $D_{\text{transform}}$ to reveal internal sub-regions representing individual object instances:

$$D_{\text{transform}}(x, y) = \min_{(i,j) \in K} \|(x, y) - (i, j)\|$$

where $K$ represents contour boundary pixels. Applying a binary threshold to $D_{\text{transform}}$ segments sub-regions within each composite contour, enabling separate identification of overlapping objects which is a challenging problem considering dense scenarios.

**Centroid Calculation for Candidate Prompts:** For each detected contour (both parent and child contours within composite regions), we calculate the centroid using spatial moments:

$$cX = \frac{M_{10}}{M_{00} + \epsilon}, \quad cY = \frac{M_{01}}{M_{00} + \epsilon}$$

where $M_{ij}$ are the moments of the contour, and $\epsilon$ is a small constant to prevent division by zero. In contour and moment analysis, $M_{00}$ is a spatial moment that represents the zeroth-order moment or area of a shape. $M_{10}$ and $M_{01}$ represent the first-order moments along the x and y axes, respectively.

For a given binary image or a region defined by a contour, $M_{00}$, $M_{10}$ and $M_{01}$ are computed as:

$$M_{00} = \sum_x \sum_y I(x, y)$$

$$M_{10} = \sum_x \sum_y x \cdot I(x, y)$$

$$M_{01} = \sum_x \sum_y y \cdot I(x, y)$$

where:

- $x$ and $y$ are the coordinates of each pixel within the region of interest.
- $I(x, y)$ is the pixel intensity at position $(x, y)$.

These centroids serve as candidate point prompts, accurately marking the locations of individual object instances in dense scenarios for downstream segmentation.

### B.2 POINT PROMPT SELECTION MODULE (PPSM)

The purpose of PPSM's adaptive threshold is to filter candidate points based on similarity scores, adjusting for object density. This threshold dynamically changes to balance inclusion of true positives while filtering out false positives in dense scenes. For better understanding, we statistically model the adaptive threshold in PPSM, where the threshold dynamically adjusts according to object count using a fixed scaling factor.

**Defining the Adaptive Threshold:** Let the cosine similarity scores $S(x, y)$ (support vs query) at each pixel position $(x, y)$ form a distribution with the maximum similarity denoted by $S_{\text{max}}$.

For simplicity, we assume that similarity scores across points can be approximated by a Gaussian distribution, with mean $\mu$ and variance $\sigma^2$. The maximum similarity $S_{\max}$ is then considered the peak or upper bound of this distribution, representing the point with the highest alignment to the target feature. The adaptive threshold $T$ for point selection is defined as:

$$T = \frac{S_{\max}\sqrt{2}}{C}$$

where $C$ represents the object count in the scene. As $C$ increases, the threshold $T$ decreases, which allows for a more inclusive selection of points when there is a higher density of objects. We chose $\sqrt{2}$ as scaling factor based on empirical results as discussed in section 5.1.

**Probability of Selecting a Point with Similarity Above Threshold:** Assuming similarity scores $S$ follow a Gaussian distribution $S \sim \mathcal{N}(\mu, \sigma^2)$, the probability $P$ of a randomly selected point having a similarity score above $T$ is:

$$P(S \geq T) = 1 - \Phi\left(\frac{T - \mu}{\sigma}\right)$$

where $\Phi$ is the cumulative distribution function (CDF) of the standard normal distribution. Substituting for $T$, we get:

$$P(S \geq T) = 1 - \Phi\left(\frac{\frac{S_{\max}\sqrt{2}}{C} - \mu}{\sigma}\right)$$

This probability increases as $C$ grows, implying that a higher object count allows for more points to meet the threshold.

**Statistical Balance of True Positives and False Positives:** For high values of $C$, the threshold $T$ approaches a smaller value, close to zero. This scaling ensures that PPSM remains inclusive in dense scenes, effectively increasing recall by accepting more points with lower similarity scores. Conversely, for smaller values of $C$, $T$ is higher, allowing only points with high similarity scores to pass the threshold. This behavior enhances precision, as fewer points are selected, with a stronger emphasis on high similarity. By dynamically adjusting $T$ with $\frac{S_{\max}\sqrt{2}}{C}$, the adaptive threshold statistically balances true positives and false positives.

## C  ADDITIONAL EXPERIMENTS AND ANALYSIS

In addition to PerSense-D, we evaluate our method on COCO-20[i] (Nguyen & Todorovic, 2019) and LVIS-92[i] (Gupta et al., 2019), following Matcher (Liu et al., 2024) data preprocessing and evaluation protocols (Table 3).

**Comparison with Methods Involving In-Domain Training:** To provide a broader perspective, we compare PerSense with both in-domain training methods and training-free approaches. Despite being a training-free framework, PerSense achieves performance comparable to several well-known in-domain training methods, as shown in Table 3 .

**Comparison with C3Det:** C3Det (Lee et al., 2022) is an interactive framework designed to provide bounding boxes for tens or hundreds of tiny objects of a specific class within a given image, based on a single user-provided click on the object of interest. To ensure a fair comparison with our training-free setup, we evaluated the performance of C3Det on the PerSense-D dataset by conducting a cross-dataset generalization test. Specifically, we utilized the C3Det model trained on Tiny-DOTA and assessed its performance on the PerSense-D dataset. The positive location prior in PerSense was used as the initial user input for C3Det to detect similar instances, and the detections were subsequently passed to SAM for segmentation. The performance comparison is summarized in Table 3, where PerSense outperformed C3Det by +23.01% mIoU. This result aligns with the performance trends reported by C3Det on the Tiny-DOTA and LCell datasets. As shown in Figure 6 of Lee et al. (2022) , with a single click, the mAP is approximately 63% for Tiny-DOTA and 55% for LCell dataset, calculated at an IoU threshold of 0.5. When transitioning to mIoU, these values naturally decline due to the stricter overlap requirements for segmentation tasks compared to detection tasks.

**Comparison with PerSAM (Point-Based Prompt Method)** PerSense consistently outperforms the recent training-free, point-based PerSAM (Zhang et al., 2024) in segmentation tasks across both sparse datasets (COCO-20[i] and LVIS-92[i]) and the dense dataset (PerSense-D). PerSense demonstrates significant improvements over PerSAM-F, achieving mIoU gains of +25.5% on COCO-20[i], +13.4% on LVIS-92[i], and +42.2% on PerSense-D dataset.

**Comparison with Matcher (Patch-Level and Box-Based Prompt Method):** Matcher (Liu et al., 2024) achieves superior performance compared to PerSense on sparse datasets like COCO-20[i] and LVIS-92[i]. This increase is due to its reliance on bidirectional patch-level feature matching and bounding box-based prompts which effectively identify distinct object regions in scenarios where objects are sparse and well-separated. In contrast, Matcher struggles with dense images in the PerSense-D dataset due to its reliance on bounding box-based prompts and its relatively limited instance-level matching capabilities, which hinders its performance when segmenting densely packed objects. PerSense outperforms Matcher by +8.8% in dense scenarios. This highlights a trade-off between point prompts and bounding box prompts in segmentation performance across sparse and dense images.

For sparse images, bounding box prompts are more effective as they encapsulate the entire object, providing more comprehensive information compared to a localized point prompt. However, as discussed in sec 1 of the paper, bounding boxes face inherent limitations in dense images due to their fixed shape, inability to effectively address occlusions, and challenges in accommodating object orientation. In such scenarios, point prompts provide superior accuracy, finer control, and greater adaptability, making them more effective in handling occlusions, clutter, and densely packed instances. For this reason, PerSense proposes the automatic generation of precise instance-level point prompts leveraging density maps, rather than relying on bounding box-based prompts.

**Comparison with SegGPT and Painter:** PerSense outperforms SegGPT (Wang et al., 2023b) on both LVIS-92[i] and PerSense-D, achieving a higher mIoU by +7.1% and +16.11%, respectively. However, SegGPT demonstrates superior performance on COCO-20[i], likely due to the inclusion of the COCO dataset in its training set. Additionally, PerSense surpasses Painter (Wang et al., 2023a) on COCO-20[i] and LVIS-92[i] by +15.9% and +15.2% mIoU, respectively, despite Painter having the COCO dataset as part of its training data.

**Additional Comments on PerSense (Sparse vs Dense Images):** PerSense generates point prompts using density maps, which are designed to emphasize the spatial distribution of densely packed objects. On sparse datasets with low object counts, the generated density map often spreads across the entire object. For instance, in the case of a single object, the density map becomes a localized spread concentrated on that object. While this allows PerSense to generate multiple point prompts for the object, it undermines the primary purpose of density maps, which is to capture variations in object density across an image.

In such scenarios, density maps provide limited utility, and simpler bounding box-based approaches prove to be more effective. In summary, while PerSense performs reasonably well on sparse datasets like COCO-20[i] and LVIS-92[i], generating density maps for sparse scenarios (small object count) is less efficient. These cases can be effectively handled by bounding box-based methods, whereas PerSense is specifically designed to excel in dense scenarios by generating precise point prompts where bounding box-based approaches often struggle.

# D  ADDITIONAL ABLATIONS

**Multiple Iterations in Feedback Mechanism:** The feedback mechanism in PerSense utilizes the initial segmentation output from the decoder to select multiple exemplars for refining the density map via DMG. This process occurs in a single pass, with exemplars selected based on their SAM scores, and does not involve multiple iterations, effectively fixing the iteration count at one. An ablation study, presented in Table 4, examines the effect of multiple iterations in feedback mechanism on segmentation accuracy as well as computational efficiency. The results indicate that additional iterations are unnecessary, as they do not improve segmentation accuracy beyond the results achieved in the single pass but instead increase computational overhead, reducing the efficiency of the PerSense pipeline. Intuitively, this is because the first-pass exemplars (four in our case) correspond to the most confident instances of the target object category. These exemplars are easily detected by DMG, with

Table 3: Comparison of PerSense with other methods on COCO-20[i], LVIS-92[i], and PerSense-D datasets.

| Methods | Venue | COCO-20[i] | | | | | LVIS-92[i] | PerSense-D |
| | | F0 | F1 | F2 | F3 | Mean mIoU | Mean mIoU | mIoU |
|---|---|---|---|---|---|---|---|---|
| *In-domain training* | | | | | | | | |
| HSNet (Min et al., 2021) | CVPR 21 | 37.2 | 44.1 | 42.4 | 41.3 | 41.2 | 17.4 | - |
| VAT (Hong et al., 2022) | ECCV 22 | 39.0 | 43.8 | 42.6 | 39.7 | 41.3 | 18.5 | - |
| FPTrans (Zhang et al., 2022) | NIPS 22 | 44.4 | 48.9 | 50.6 | 44.0 | 47.0 | - | - |
| MIANet (Yang et al., 2023) | CVPR 23 | 42.4 | 52.9 | 47.7 | 47.4 | 47.6 | - | - |
| LLaFS (Zhu et al., 2024) | CVPR 24 | 47.5 | 58.8 | 56.2 | 53.0 | 53.9 | - | - |
| *COCO as training data* | | | | | | | | |
| Painter (Wang et al., 2023a) | CVPR 23 | 31.2 | 35.3 | 33.5 | 32.4 | 33.1 | 10.5 | - |
| SegGPT (Wang et al., 2023b) | ICCV 23 | 56.3 | 57.4 | 58.9 | 51.7 | 56.1 | 18.6 | 55.5 |
| *Tiny-DOTA as training data* | | | | | | | | |
| C3Det (Lee et al., 2022) | CVPR 22 | - | - | - | - | - | - | 48.6 |
| *Training-free* | | | | | | | | |
| PerSAM (Zhang et al., 2024) | | 23.1 | 23.6 | 22.0 | 23.4 | 23.0 | 11.5 | 24.4 |
| PerSAM-F (Zhang et al., 2024) | ICLR 24 | 22.3 | 24.0 | 23.4 | 24.1 | 23.5 | 12.3 | 29.3 |
| Matcher (Liu et al., 2024) | | 52.7 | 53.5 | 52.6 | 52.1 | 52.7 | 33.0 | 62.8 |
| PerSense | (this work) | 47.8 | 49.3 | 48.9 | 50.1 | 49.0 | 25.7 | 71.6 |

Table 4: Impact of multiple feedback mechanism iterations on PerSense performance.

| No. of iterations (Feedback Mechanism) | PerSense mIoU | Average inference time per image (sec) |
|---|---|---|
| 1 | 71.61 | 2.7 |
| 2 | 71.65 | 3.1 |
| 3 | 71.63 | 3.5 |
| 4 | 71.60 | 3.9 |

their boundaries well delineated by SAM, even when using a single initially selected exemplar as input. In subsequent iterations, the same exemplars are repeatedly selected due to their distinct visual features and consistently high SAM scores, attributed to the clearly defined boundaries in their segmentation masks. Consequently, multiple feedback iterations provide no additional benefit, rendering further iterations redundant.

**Component-wise Ablation Study of PerSense on COCO dataset:** As suggested, in addition to the PerSense-D dataset, we provide a component-wise ablation study of PerSense on the COCO dataset in Table 5. The results demonstrate that integrating PPSM into the proposed baseline leads to a +2.48% mIoU improvement, as it effectively filters out false positives from the candidate point prompts generated by IDM. On the other hand, the feedback mechanism yields a modest +0.19% mIoU improvement, which is expected for images with a low object count. For example, if an image contains only a single object instance, the feedback mechanism cannot select four exemplars, limiting its ability to further refine the initial density map.

**Running Efficiency Comparison:** Alongside the inference time comparison presented in Table 1, we also provide details on memory consumption for PerSense, evaluated on a single NVIDIA GeForce RTX 4090 GPU with a batch size of 1 (Table 6). PerSense is highly computationally efficient than Matcher and PerSAM-F and incurs marginal latency and GPU memory usage compared to Grounded-SAM.

Table 5: Component-wise ablation study of PerSense on COCO dataset.

| Method | IDM | PPSM | Feedback | DMG: DSALVANet mIoU(Gain) |
|---|---|---|---|---|
| proposed baseline | yes | no | no | 46.33 (-) |
| proposed baseline + PPSM | yes | yes | no | 48.81 (+2.48) |
| PerSense | yes | yes | yes | 49.00 (+0.19) |

Table 6: Running efficiency comparison of PerSense with SOTA.

| Method | Memory (MB) | Avg inference time (per image) (sec) |
|---|---|---|
| Grounded-SAM (Ren et al., 2024) | 2943 | 1.8 |
| PerSAM-F (Zhang et al., 2024) | 2950 | 47.78 |
| Matcher (Liu et al., 2024) | 3209 | 10.2 |
| PerSense (this work) | 2988 | 2.7 |

