# OpenReview forum: "PerSense: Personalized Instance Segmentation in Dense Images"
_ICLR.cc/2025/Conference — Submitted to ICLR 2025_

### Official Review · Reviewer_L45P · 2024-11-01

**Soundness:** 4
**Presentation:** 4
**Contribution:** 3
**Rating:** 6
**Confidence:** 4

**Summary:**

This paper proposes a training-free and model-agnostic one-shot framework for personalized instance segmentation in dense images. The proposed framework, PerSense, is featured with a Instance Detection Module (IDM), a Point Prompt Selection Module (PPSM), and feedback machanism to improve the accuracy of density maps. Besides the framework, a dataset PerSense-D is introduced for personalized segmentation in dense images and demonstrating the superiority of the proposed method by comparing it with the SOTA on this benchmark. According to the quantitative and qualitative results, the proposed method shows a performance improvement with a clear margin. Experiments are well analyzed and ablation study is well designed to show the effectiveness of proposed component.

**Strengths:**

1. Personalized instance-level segmentation which this paper target to solve is an interesting and important scenario for automation. The training-free manner ensures the ease for deployment.
2. Good performance of the proposed method from both qualitative and quantitative view and well experimental analysis.
3. Dataset contribution. PerSense-D is introduced as a dataset exclusive to personalized segmentation in dense images.
4. The whole paper is well organized and presented.

**Weaknesses:**

1. This paper claim they provide a one-shot personalized segmentation framework in instance level. However, the main results reported in table 1 seems come from 4-shot setting according to table 2 (c). If so, the 1-shot result should be reported to support the claimed contribution.
2. The proposed framework is complex by combining lots of existing methods or modules. Although the inference time is evaluated, memory consumption is not reported, which is critical for real deployment.
3. No result on standard segmentation datasets is reported. Results on datasets with fewer objects can provide a better understanding on the proposed framework.
4. It seems the feedback mechanism plays an important role to ensure a high performance. But better feedback may involve more local iteration to avoid false positive and instance missing, which leads to longer inference time.

**Questions:**

1. Will the codes and PreSense-D dataset be released?
2. Why does PerSense fail to have a superior performance over other methods on standard segmentation datasets with few object instances? Could authors have more discussion on the drawbacks of the proposed framework?

---

> ### Author Response · Authors · 2024-11-22
> **Response to Reviewer L45P (Part # 1)**
>
> We sincerely appreciate your constructive feedback. We hope our response can address your concerns.
>
> >**Q1: The main results reported in table 1 seems come from 4-shot setting according to table 2 (c). If so, the 1-shot result should be reported to support the claimed contribution.**
>
> The results in Table 2(c) pertain exclusively to the feedback mechanism, where we provide performance ablations for PerSense using different numbers of exemplars for the Density Map Generator (DMG). The feedback mechanism selects exemplars from the initial segmentation output of PerSense, which is generated using the one-shot support set (a reference image and its corresponding mask). Consequently, all results presented in Table 2(c) adhere to the one-shot setting, as the feedback mechanism operates solely on exemplars derived from the segmentation output generated with the one-shot support set. This table highlights the impact of selecting varying numbers of exemplars from the initial segmentation output, thereby justifying our choice of four exemplars in the feedback mechanism to refine the initial density maps effectively. For further clarity, please refer to Figure 7 in Appendix A.2 of the paper, which provides component-wise qualitative analysis of PerSense, including the functionality of the feedback mechanism. This figure shall assist in better understanding that how the feedback mechanism selects exemplars from the intermediate one-shot segmentation output in PerSense.
>
> >**Q2: Although the inference time is evaluated, memory consumption is not reported, which is critical for real deployment.**
>
> Alongside the inference time comparison already presented in Table 1 of the paper, we also provide details on memory consumption for PerSense, evaluated on a single NVIDIA GeForce RTX 4090 GPU with a batch size of 1. PerSense is highly computationally efficient than Matcher [1] and PerSAM-F [2] and incurs marginal latency and GPU memory usage compared to Grounded-SAM [3].
>
> |       Method        | Memory (MB) | Avg Inference time (per image) (sec) |
> |-------------------|:-----------:|:------------------------------------:|
> |   Grounded-SAM [3]      |    2943     |                1.8                  |
> |     PerSAM-F [2]       |    2950     |               47.78                 |
> |       Matcher [1]      |    3209     |               10.2                  |
> |      PerSense       |    2988     |                2.7                  |
>
> ---
>
> >**Q3: No result on standard segmentation datasets is reported. Results on datasets with fewer objects can provide a better understanding on the proposed framework.**
>
> In addition to PerSense-D, we evaluate our method on standard segmentation datasets, namely, COCO-20$^i$ [4] and LVIS-92$^i$ [5], following Matcher [1] data preprocessing and evaluation protocols. Please refer to the table below. We have also included these additional evaluations and comparisons as Appendix C in the updated pdf.
>
> **Comparison with methods involving in-domain training:**  To provide a broader perspective, we compare PerSense with both in-domain training methods and training-free approaches. Despite being a training-free framework, PerSense achieves performance comparable to several well-known in-domain training methods as shown in the table below.
>
> **Comparison with C3Det [6]:**  C3Det is an interactive framework designed to provide bounding boxes for tens or hundreds of tiny objects of a specific class within a given image, based on a single user-provided click on the object of interest. To ensure a fair comparison with our training-free setup, we evaluated the performance of C3Det on the PerSense-D dataset by conducting a cross-dataset generalization test. Specifically, we utilized the C3Det model trained on Tiny-DOTA and assessed its performance on the PerSense-D dataset. The positive location prior in PerSense was used as the initial user input for C3Det to detect similar instances, and the detections were subsequently passed to SAM for segmentation. The performance comparison is summarized in the table below, where PerSense outperformed C3Det by +23.01% mIoU.
>
> **Comparison with PerSAM [2] (point-based prompt method):**  PerSense consistently outperforms the recent training-free, point-based PerSAM in segmentation tasks across both sparse datasets (COCO-20$^i$ and LVIS-92$^i$) and the dense dataset (PerSense-D). PerSense demonstrates significant improvements over PerSAM-F, achieving mIoU gains of +25.5% on COCO-20$^i$, +13.4% on LVIS-92$^i$, and +42.2% on PerSense-D.
>
> **CONTD..**
>
> ---
>
> References:
>
> [1] Matcher: Segment anything with one shot using all-purpose feature matching. ICLR 2024.
>
> [2] Personalize Segment Anything Model with one shot. ICLR 2024.
>
> [3] Grounded sam: Assembling open-world models for diverse visual tasks. arXiv 2024.
>
> [4] Feature weighting for FSS. ICCV 2019
>
> [5] Dataset for large vocabulary instance segmentation. CVPR 2019
>
> [6] Interactive multi-class tiny-object detection. CVPR 2022.

---

> ### Author Response · Authors · 2024-11-22
> **Response to Reviewer L45P (Part # 2)**
>
> >** Q3 contd...**
>
> **Comparison with Matcher [1] (patch-level and box-based prompt method):**  Matcher achieves superior performance than PerSense on sparse datasets like COCO-20$^i$ and LVIS-92$^i$. This increase is due to its reliance on bidirectional patch-level feature matching and bounding box-based prompts which effectively identify distinct object regions in scenarios where objects are sparse and well-separated. In contrast, Matcher struggles with dense images in the PerSense-D dataset due to its reliance on bounding box-based prompts and its relatively limited instance-level matching capabilities, which hinders its performance when segmenting densely packed objects. PerSense outperforms Matcher by +8.8%. This highlights a trade-off between point prompts and bounding box prompts in segmentation performance across sparse and dense images. For sparse images, bounding box prompts are more effective as they encapsulate the entire object, providing more comprehensive information compared to a localized point prompt. However, as discussed in Section 1 (L: 84–89) of the paper, bounding boxes face inherent limitations in dense images due to their fixed shape, inability to effectively address occlusions, and challenges in accommodating object orientation. In such scenarios, point prompts provide superior accuracy, finer control, and greater adaptability, making them more effective in handling occlusions, clutter, and densely packed instances. For this reason, PerSense proposes the automatic generation of precise instance-level point prompts leveraging density maps, rather than relying on bounding box-based prompts.
>
> **Comparison with SegGPT [7] and Painter [8]:**  PerSense outperforms SegGPT on both LVIS-92$^i$ and PerSense-D, achieving a higher mIoU by +7.1% and +16.11%, respectively. However, SegGPT demonstrates superior performance on COCO-20$^i$, likely due to the inclusion of the COCO dataset in its training set. Additionally, PerSense surpasses Painter on COCO-20$^i$ and LVIS-92$^i$ by +15.9% and +15.2% mIoU, respectively, despite Painter having the COCO dataset as part of its training data.
>
> ---
>
> | Methods| Venue| COCO-20$^i$ F0| COCO-20$^i$ F1| COCO-20$^i$ F2| COCO-20$^i$ F3| Mean mIoU| LVIS-92$^i$ mIoU| PerSense-D mIoU|
> |---------------|---------------------|----------------|----------------|----------------|----------------|-----------|------------------|-----------------|
> | **In-domain training**|||||||||
> | HSNet [9]|ICCV 21|37.2|44.1|42.4|41.3|41.2|17.4|-|
> | VAT [10]|ECCV 22|39.0|43.8|42.6|39.7|41.3|18.5|-|
> | FPTrans [11]|NeurIPS 22|44.4|48.9|50.6|44.0|47.0|-|-|
> | MIANet [12]|CVPR 23|42.4|52.9|47.7|47.4|47.6|-|-|
> | LLaFS [13]|CVPR 24|47.5|58.8|56.2|53.0|53.9|-|-|
> | **COCO included in training data**|||||||||
> | Painter [8]|CVPR 23|31.2|35.3|33.5|32.4|33.1|10.5|-|
> | SegGPT [7]|ICCV 23|56.3|57.4|58.9|51.7|56.1|18.6|55.5|
> | **Tiny-DOTA as training data**|||||||||
> | C3Det [6]|CVPR 2022|-|-|-|-|-|-|48.6|
> | **Training-free**|||||||||
> | PerSAM [2]|ICLR 24|23.1|23.6|22.0|23.4|23.0|11.5|24.4|
> | PerSAM-F [2]|ICLR 24|22.3|24.0|23.4|24.1|23.5|12.3|29.3|
> | Matcher [1]|-|52.7|53.5|52.6|52.1|52.7|33.0|62.8|
> | PerSense|(this work)|47.8|49.3|48.9|50.1|49.0|25.7|71.6|
>
>
> ---
>
> References:
>
> [7] Seggpt: Segmenting everything in context. ICCV 2023.
>
> [8] Images speak in images: A generalist painter for in-context visual learning. CVPR 2023.
>
> [9] Hypercorrelation squeeze for few-shot segmentation. ICCV 2021.
>
> [10] Cost aggregation with 4d convolutional swin transformer for few-shot segmentation. ECCV 2022.
>
> [11] Feature-proxy transformer for few-shot segmentation. NeurIPS 2022.
>
> [12] Aggregating unbiased instance information for few-shot semantic segmentation. CVPR 2023.
>
> [13] When large language models meet few-shot segmentation. CVPR 2024.

---

> ### Author Response · Authors · 2024-11-22
> **Response to Reviewer L45P (Part # 3)**
>
> >**Q4: It seems the feedback mechanism plays an important role to ensure a high performance. But better feedback may involve more local iteration to avoid false positive and instance missing, which leads to longer inference time.**
>
> The feedback mechanism in PerSense leverages the initial segmentation output from the decoder to select multiple exemplars for refining the density map via the DMG. This process is designed to occur in a single pass, where exemplars are selected based on their SAM scores. Importantly, the mechanism operates without requiring multiple iterations, effectively fixing the iteration count at one and thereby introducing no additional inference time overhead. An ablation study, presented below, examines the effect of multiple iterations in feedback mechanism on segmentation accuracy as well as computational efficiency. The results indicate that additional iterations are unnecessary, as they do not improve segmentation accuracy beyond the results achieved in the single pass but instead increase computational overhead, reducing the efficiency of the PerSense pipeline. Intuitively, this is because the first-pass exemplars (four in our case) correspond to the most confident instances of the target object category. These exemplars are easily detected by DMG, with their boundaries well delineated by SAM, even when using a single initially selected exemplar as input. In subsequent iterations, the same exemplars are repeatedly selected due to their distinct visual features and consistently high SAM scores, attributed to the clearly defined boundaries in their segmentation masks. Consequently, multiple feedback iterations provide no additional benefit, rendering further iterations redundant.
>
> ---
>
> | No. of iterations (Feedback Mechanism) | PerSense mIoU | Average Inference time per image (sec) |
> |:--------------------------------------:|:-------------:|:-------------------------------------:|
> |                   1                    |     71.61     |                 2.7                  |
> |                   2                    |     71.65     |                 3.1                  |
> |                   3                    |     71.63     |                 3.5                  |
> |                   4                    |     71.60     |                 3.9                  |
>
> ---
>
> >**Q5: Will the codes and PerSense-D dataset be released?**
>
> Yes, we aim to publicly release the code and dataset after the final decision. Please see supplementary material for further details.
>
> ---
>
> >**Q6: Why does PerSense fail to have a superior performance over other methods on standard segmentation datasets with few object instances? Could authors have more discussion on the drawbacks of the proposed framework?**
>
> We do not characterize PerSense as failing to achieve superior performance over all other methods on standard segmentation datasets with fewer object instances. Instead, we emphasize the fairness of such comparisons given the distinct focus of PerSense on dense instance segmentation. For a detailed performance analysis of PerSense on standard segmentation datasets, please refer to the response to Q3.
>
> PerSense generates point prompts using density maps, which are designed to emphasize the spatial distribution of densely packed objects. On sparse datasets with low object counts, the generated density map often spreads across the entire object. For instance, in the case of a single object, the density map becomes a localized spread concentrated on that object. While this allows PerSense to generate multiple point prompts for the object, it undermines the primary purpose of density maps, which is to capture variations in object density across an image. In such scenarios, density maps provide limited utility, and simpler bounding box-based approaches prove to be more effective.
>
> In summary, while PerSense performs reasonably well on sparse datasets like COCO-20$^i$ and LVIS-92$^i$, generating density maps for sparse scenarios (small object count) is less efficient. These cases can be effectively handled by bounding box-based methods, whereas PerSense is specifically designed to excel in dense scenarios by generating precise point prompts where bounding box-based approaches often fail.
>
> ---

---

> > ### Comment · Reviewer_L45P · 2024-11-26
> > **Response to the rebuttal**
> >
> > The rebuttal addresses most of my concerns, especially the misunderstanding in Q1. Therefore, I will maintain my rating.

---

> > > ### Author Response · Authors · 2024-11-26
> > > **Thanks for your feedback!**
> > >
> > > Thank you for acknowledging our rebuttal and efforts.

---

### Official Review · Reviewer_GosF · 2024-11-01

**Soundness:** 2
**Presentation:** 2
**Contribution:** 2
**Rating:** 5
**Confidence:** 3

**Summary:**

This work focuses on personalized instance segmentation in dense and crowded scenarios. To this end, they propose PerSense, a training-free framework. Specifically, they use a class-label extractor (CLE) and a grounding detector to select the effective exemplars for a density map generator (DMG). Then, they propose an instance detection module (IDM) to generate point prompts from the density map and design a point prompt selection module (PPSM) to reduce false positive predictions. Last, the authors further introduce a feedback scheme to refine the exemplar selection and thus improve the performance of PerSense. Moreover, they propose a dataset, PerSense-D, for the evaluation of instance segmentation in dense images. However, I still have some questions about the experimental settings.

**Strengths:**

1. The proposed PPSM effectively removes the false positive predictions.
2. The proposed feedback mechanism further improves the density maps of DMG and boosts the performance of personalized instance segmentation.
3. The authors propose a new data set with 717 images and 28,395 objects for personalized instance segmentation.

**Weaknesses:**

1. The authors compare the proposed method with the SOTA methods on PerSense-D. However, I noticed that the authors conducted ablation studies on the same dataset to pursue the best performance. As there are no more results on other datasets, the superior performance of the proposed method needs to be verified.
2. In Table 2, it is unclear why the method achieves the best performance with a normalized factor equal to sqrt(2). More discussions on the insight of normalized factors are required.
3. Although the feedback mechanism effectively improves the performance of the proposed method, it seems that this mechanism violates the one-shot setting. In this sense, the comparison with Grounded-SAM seems not fair.
4. At line 379, page 8, “x 1.02” should be “$\times 1.02$”.
5. In Table 2(c), “No of Shots” should be “No. of Shots”.

**Questions:**

I think at least the authors should split the proposed dataset for validation and testing.

---

> ### Author Response · Authors · 2024-11-22
> **Response to Reviewer GosF (Part # 1)**
>
> We sincerely appreciate your constructive feedback. We hope our response can address your concerns.
>
> >**Q1:  The authors compare the proposed method with the SOTA methods on PerSense-D. However, I noticed that the authors conducted ablation studies on the same dataset to pursue the best performance. As there are no more results on other datasets, the superior performance of the proposed method needs to be verified.**
>
>
> **Component-wise Ablation Study of PerSense on COCO dataset [1]:**  As suggested, in addition to the PerSense-D dataset, we provide a component-wise ablation study of PerSense on the COCO dataset in the table below. The results demonstrate that integrating PPSM into the proposed baseline leads to a +2.48% mIoU improvement, as it effectively filters out false positives from the candidate point prompts generated by IDM. On the other hand, the feedback mechanism yields a modest +0.19% mIoU improvement, which is expected for images with a low object count. For example, if an image contains only a single object instance, the feedback mechanism cannot select four exemplars, limiting its ability to further refine the initial density map. We have also added this ablation study in Appendix C of the updated pdf.
>
> | Method                         | IDM | PPSM | Feedback | Mean mIoU (Gain) |
> |--------------------------------|:-----:|:------:|:----------:|:------------------:|
> | proposed baseline              | yes | no   | no       | 46.33 (-)        |
> | proposed baseline + PPSM       | yes | yes  | no       | 48.81 (+2.48)    |
> | PerSense                       | yes | yes  | yes      | 49.00 (+0.19)    |
>
> **Evaluation on Additional Datasets:**  In addition to PerSense-D, we evaluate our method on COCO-20$^i$ [1] and LVIS-92$^i$ [2], following Matcher [3] data preprocessing and evaluation protocols. Please refer to the table below. We have also included these additional evaluations and comparisons as Appendix C in the updated pdf.
>
> **Comparison with methods involving in-domain training:**  To provide a broader perspective, we compare PerSense with both in-domain training methods and training-free approaches. Despite being a training-free framework, PerSense achieves performance comparable to several well-known in-domain training methods as shown in the table below.
>
> **Comparison with C3Det [4]:**  C3Det is an interactive framework designed to provide bounding boxes for tens or hundreds of tiny objects of a specific class within a given image, based on a single user-provided click on the object of interest. To ensure a fair comparison with our training-free setup, we evaluated the performance of C3Det on the PerSense-D dataset by conducting a cross-dataset generalization test. Specifically, we utilized the C3Det model trained on Tiny-DOTA and assessed its performance on the PerSense-D dataset. The positive location prior in PerSense was used as the initial user input for C3Det to detect similar instances, and the detections were subsequently passed to SAM for segmentation. The performance comparison is summarized in the table below, where PerSense outperformed C3Det by +23.01% mIoU.
>
> **Comparison with PerSAM [5] (point-based prompt method):**  PerSense consistently outperforms the recent training-free, point-based PerSAM in segmentation tasks across both sparse datasets (COCO-20$^i$ and LVIS-92$^i$) and the dense dataset (PerSense-D). PerSense demonstrates significant improvements over PerSAM-F, achieving mIoU gains of +25.5% on COCO-20$^i$, +13.4% on LVIS-92$^i$, and +42.2% on PerSense-D.
>
> **Comparison with SegGPT [6] and Painter [7]:**
> PerSense outperforms SegGPT on both LVIS-92$^i$ and PerSense-D, achieving a higher mIoU by +7.1% and +16.11%, respectively. However, SegGPT demonstrates superior performance on COCO-20$^i$, likely due to the inclusion of the COCO dataset in its training set. Additionally, PerSense surpasses Painter on COCO-20$^i$ and LVIS-92$^i$ by +15.9% and +15.2% mIoU, respectively, despite Painter having the COCO dataset as part of its training data.
>
> **CONTD..**
>
>
> ---
>
> References:
>
> [1] Feature weighting for few-shot segmentation. ICCV 2019
>
> [2] Dataset for large vocabulary instance segmentation. CVPR 2019
>
> [3] Matcher: Segment anything with one shot using all-purpose feature matching. ICLR 2024.
>
> [4] Interactive multi-class tiny-object detection. CVPR 2022.
>
> [5] Personalize Segment Anything Model with one shot. ICLR 2024.
>
> [6] Seggpt: Segmenting everything in context. ICCV 2023.
>
> [7] Images speak in images: A generalist painter for in-context visual learning. CVPR 2023.

---

> ### Author Response · Authors · 2024-11-22
> **Response to Reviewer GosF (Part # 2)**
>
> > **Q1 contd..**
>
> **Comparison with Matcher [7] (patch-level and box-based prompt method):**  Matcher achieves superior performance than Persense on sparse datasets like COCO-20$^i$ [1] and LVIS-92$^i$ [2]. This increase is due to its reliance on bidirectional patch-level feature matching and bounding box-based prompts which effectively identify distinct object regions in scenarios where objects are sparse and well-separated. In contrast, Matcher struggles with dense images in the PerSense-D dataset due to its reliance on bounding box-based prompts and its relatively limited instance-level matching capabilities, which hinders its performance when segmenting densely packed objects. PerSense outperforms Matcher by +8.8%. This highlights a trade-off between point prompts and bounding box prompts in segmentation performance across sparse and dense images. For sparse images, bounding box prompts are more effective as they encapsulate the entire object, providing more comprehensive information compared to a localized point prompt. However, as discussed in Section 1 (L: 84–89) of the paper, bounding boxes face inherent limitations in dense images due to their fixed shape, inability to effectively address occlusions, and challenges in accommodating object orientation. In such scenarios, point prompts provide superior accuracy, finer control, and greater adaptability, making them more effective in handling occlusions, clutter, and densely packed instances. For this reason, PerSense proposes the automatic generation of precise instance-level point prompts leveraging density maps, rather than relying on bounding box-based prompts.
>
> | Methods| Venue| COCO-20$^i$ F0| COCO-20$^i$ F1| COCO-20$^i$ F2| COCO-20$^i$ F3| Mean mIoU| LVIS-92$^i$ mIoU| PerSense-D mIoU|
> |---------------|---------------------|----------------|----------------|----------------|----------------|-----------|------------------|-----------------|
> | **In-domain training**|||||||||
> | HSNet [8]|ICCV 21|37.2|44.1|42.4|41.3|41.2|17.4|-|
> | VAT [9]|ECCV 22|39.0|43.8|42.6|39.7|41.3|18.5|-|
> | FPTrans [10]|NeurIPS 22|44.4|48.9|50.6|44.0|47.0|-|-|
> | MIANet [11]|CVPR 23|42.4|52.9|47.7|47.4|47.6|-|-|
> | LLaFS [12]|CVPR 24|47.5|58.8|56.2|53.0|53.9|-|-|
> | **COCO included in training data**|||||||||
> | Painter [7]|CVPR 23|31.2|35.3|33.5|32.4|33.1|10.5|-|
> | SegGPT [6]|ICCV 23|56.3|57.4|58.9|51.7|56.1|18.6|55.5|
> | **Tiny-DOTA as training data**|||||||||
> | C3Det [4]|CVPR 2022|-|-|-|-|-|-|48.6|
> | **Training-free**|||||||||
> | PerSAM [5]|ICLR 24|23.1|23.6|22.0|23.4|23.0|11.5|24.4|
> | PerSAM-F [5]|ICLR 24|22.3|24.0|23.4|24.1|23.5|12.3|29.3|
> | Matcher [3]|-|52.7|53.5|52.6|52.1|52.7|33.0|62.8|
> | PerSense|(this work)|47.8|49.3|48.9|50.1|49.0|25.7|71.6|
>
> ---
>
> >**Q2: In Table 2, it is unclear why the method achieves the best performance with a normalized factor equal to sqrt(2). More discussions on the insight of normalized factors are required.**
>
> We chose $\sqrt{2}$ as a scaling factor based on empirical results as discussed in section 5.1 of the paper. The purpose of PPSM’s adaptive threshold is to filter candidate points based on similarity scores, adjusting for object density. This threshold dynamically changes to balance the  inclusion of true positives while filtering out false positives in dense scenes. For better understanding, we statistically model the adaptive threshold in PPSM, where the threshold dynamically adjusts according to object count using a fixed normalization factor of $\sqrt{2}$.
>
> Let the cosine similarity scores $S(x, y)$ (support vs query) at each pixel position $(x, y)$ form a distribution with the maximum similarity denoted by $S_{\max}$. For simplicity, we assume that similarity scores across points can be approximated by a Gaussian distribution, with mean $\mu$ and variance $\sigma^2$. The maximum similarity $S_{\max}$ is then considered the peak or upper bound of this distribution, representing the point with the highest alignment to the target feature. The adaptive threshold $T$ for point selection is defined as:
>
> $$
> T = \frac{S_{\max} \sqrt{2}}{C}
> $$
>
> where $C$ represents the object count in the scene. As $C$ increases, the threshold $T$ decreases, which allows for a more inclusive selection of points when there is a higher density of objects.
>
> ---
>
> References:
>
> [8] Hypercorrelation squeeze for few-shot segmentation. ICCV 2021.
>
> [9] Cost aggregation with 4d convolutional swin transformer for few-shot segmentation. ECCV 2022.
>
> [10] Feature-proxy transformer for few-shot segmentation. NeurIPS 2022.
>
> [11] Aggregating unbiased instance information for few-shot semantic segmentation. CVPR 2023.
>
> [12] When large language models meet few-shot segmentation. CVPR 2024.

---

> ### Author Response · Authors · 2024-11-22
> **Response to Reviewer GosF (Part # 3)**
>
> >**Q3: Although the feedback mechanism effectively improves the performance of the proposed method, it seems that this mechanism violates the one-shot setting. In this sense, the comparison with Grounded-SAM seems not fair.**
>
> In PerSense, the feedback mechanism is designed to refine density maps by leveraging the intermediate segmentation results obtained from initial one-shot segmentation. This mechanism strictly adheres to the one-shot setting, as it relies solely on the intermediate segmentation output from the query image without incorporating additional labeled data or external supervision. It does not introduce any new information beyond what is provided by the one-shot support set and the intrinsic content of the query image. Serving as a stand-alone refinement strategy, the feedback mechanism innovatively utilizes the initial segmentation output generated by PerSense to enhance density map quality. Furthermore, PerSense’s Point Prompt Selection Module (PPSM) employs cosine similarity between the query and support images to determine an adaptive threshold for point prompt selection. This cosine similarity is calculated solely based on the one-shot support set and operates independently of the multiple exemplars identified by the feedback mechanism. The exemplars are used exclusively for refining the density map by drawing from the information embedded in the initial segmentation output, ensuring that PerSense remains fully compliant with the constraints of the one-shot paradigm.
>
> ---
>
> >**Q4: At line 379, page 8, “x 1.02” should be corrected. In Table 2(c), “No of Shots” should be “No. of Shots”.**
>
> Thank you for pointing this out. We have revised the updated pdf accordingly.
>
> ---
>
> >**Q5: I think at least the authors should split the proposed dataset for validation and testing.**
>
> Currently, we have positioned the PerSense-D dataset as a challenging evaluation benchmark for one-shot personalized segmentation methods in dense scenarios. Our objective was to highlight that even with the advanced prompt-engineering capabilities of vision foundational models, extensively trained on large-scale segmentation datasets, state-of-the-art methods face significant challenges. These challenges persist in achieving effective, fully automatic personalized segmentation (without user intervention) under the one-shot setting in dense environments. Moving forward, we plan to extend PerSense-D dataset by introducing dedicated training, validation, and test splits to facilitate the development and training of specialized models tailored for this task.
>
> ---

---

> ### Comment · Reviewer_GosF · 2024-11-25
>
> Thanks for the detailed response. I still have some concerns:
> 1. As mentioned by the authors, there is no dataset for dense image segmentation, and they released persense-D. However, a proper evaluation scheme is also important when releasing a new dataset. Existing work like COCO and LVIS provides k-fold evaluation or extra validation set for fair comparisons. Following the evaluation scheme on the persense-D, I think it may result in the risk of overfitting. Besides, it is unhealthy to pursue the best performance on the same dataset and compare it with other methods (for this work or the following studies that use the proposed persense-D).
>
> Overall, the absence of a proper evaluation scheme on persense-D reduces the contributions of this work.
>
> 2. In Table 2, I think it is a common practice to use values like $1,2,3...$. I'm curious whether the values of $1, \sqrt{2}, \sqrt{3}, \sqrt{5}...$ are related to the $\sigma^2$.

---

> > ### Author Response · Authors · 2024-11-29
> > **Happy to Provide Additional Clarifications!**
> >
> > We sincerely appreciate your thoughtful and thorough review of our paper. We have carefully addressed each of your points and greatly value your constructive feedback, which has contributed to the improvement of our work. As the Author-Reviewer discussion period is nearing its conclusion, please feel free to share any additional concerns or suggestions—we would be more than happy to address them. Thank you.

---

> ### Author Response · Authors · 2024-11-25
> **Response to Reviewer GosF**
>
> Thank you for your valuable feedback and acknowledging our efforts. We hope that our response address your concerns.
>
> >**Q.1: As mentioned by the authors, there is no dataset for dense image segmentation, and they released persense-D. However, a proper evaluation scheme is also important when releasing a new dataset. Existing work like COCO and LVIS provides k-fold evaluation or extra validation set for fair comparisons. Following the evaluation scheme on the persense-D, I think it may result in the risk of overfitting. Besides, it is unhealthy to pursue the best performance on the same dataset and compare it with other methods (for this work or the following studies that use the proposed persense-D).**
>
> We take the opportunity to clarify that the proposed PerSense framework is entirely training-free, and the PerSense-D dataset serves exclusively as a test set designed to evaluate challenging scenarios of personalized segmentation in dense images under a one-shot setting. Unlike the k-fold validation in COCO and LVIS, which involves iterative training and testing to reduce variance caused by a single train-test split, none of the evaluation experiments reported in our paper involve training of PerSense on PerSense-D dataset. The entire PerSense-D dataset is proposed and utilized solely as an evaluation benchmark. As outlined in Section 4 (L: 317–320) of the paper, to ensure fair comparisons across various one-shot approaches and eliminate variability due to random seeding, we explicitly provide 28 support / reference images—one for each of the 28 object categories in PerSense-D. Each reference image contains a single object instance along with its mask, intended to guide personalized segmentation for its corresponding object category. This design ensures consistency, reproducibility, and fairness in the evaluation process.
>
> In the future, we plan to further extend the scale of PerSense-D dataset by introducing dedicated training, validation, and test splits to facilitate the development and training of specialized models for personalized instance segmentation in dense images.
>
> ---
>
> >**Q2: In Table 2, I think it is a common practice to use values like  $ 1 $, $ 2 $, $ 3 $. . . I'm curious whether the values of  $ 1 $, $ \sqrt{2} $, $ \sqrt{3} $ $ \sqrt{5} $...  are related to the $ \sigma^2 $.**
>
> As discussed in Section 5.2 of the paper, the choice of $ \sqrt{2} $ as a scaling factor is primarily based on empirical results. Starting with an initial value of $ 1 $, we adopted a square root progression to achieve a more gradual increase / step-size compared to linear progression (e.g., $ 1 $, $ 2 $, $ 3 $). Linear progression increases the threshold too aggressively for higher object counts, leading to overly strict thresholds that reject valid point prompts. This decline in performance is evident in Table 2(b), which shows decline in performance with a scaling factor of $ \sqrt{5} \ (2.23)$.
>
> Additionally, the selection of this scaling factor is independent of the distribution of similarity scores. Instead, it regulates the adaptability of the point prompt selection criterion in PPSM, which is determined by the deterministic count of objects in an image. Ultimately, $ \sqrt{2} $ serves as an empirically optimized factor that balances sensitivity and robustness in point prompt selection, ensuring consistent performance across varying object densities.
>
> ---

---

### Official Review · Reviewer_CscQ · 2024-11-03

**Soundness:** 2
**Presentation:** 2
**Contribution:** 3
**Rating:** 6
**Confidence:** 5

**Summary:**

The paper proposes a training-free method to achieve personalized instance segmentation, which aims to segment what people want with refering images.  It develops a new baseline capable of automatically generating instance-level point prompts via proposing a novel Instance Detection Module (IDM) that leverages density maps, encapsulating spatial distribution of objects in an imageA dataset PerSense-D is proposed to boost this research area.

**Strengths:**

1. This article's research on the task of refining segmentation is valuable. The proposed dataset is helpful for the development of this field.
2. This method can achieve the desired effect without training, and has higher application value in this regard.

**Weaknesses:**

1. The writing and organization of the article need to be improved:
a. In Figure 1, "(a)" appears twice
b. The method section lacks a paragraph that links together how several modules operate. And the caption in Figure 2 is also very brief, requiring a description of the structure.
c. It is not recommended to write the abstract in the form of contribution (1), (2), (3)
2. In line 43, I did not get the difference between personalized instance segmentation and traditional instance segmentation. The traditional setting is also aimed at segmenting the specified categories in the image. Do I need to give a category name for input? After reading the article, my understanding is that by providing a template for referencing, other similar targets are required to be segmented. It does not mean segmenting the specified category, but segmenting the specified referencing. If I understand correctly, C3Det([1],CVPR2022) has a similar idea (even if it is a detection) and needs to be compared and discussed (its inspection results can be used for segmentation by SAM). SegGPT([2] ICCV2023) also uses some template images to segment the desired objects. Please compare and discuss.
3. SAPNet (CVPR2024) uses Point prompt combined with SAM to generate candidate masks, and selects masks that meet specific categories as outputs. This idea is similar to this article, please discuss it. (Even if the method uses point annotation, the point prompt predicted in the first stage of this article can be used as its input instead of point annotation.) In addition, methods such as Bestie (CVPR2022) have also predicted peak points as point prompts through affinity maps, density maps, and other methods. Please compare and discuss.
4. Although COCO and LVIS are not specifically designed for dense scenarios, they do not hinder the validation of our method in this paper. And the size of COCO and LVIS is still much larger than the dataset in this article. I acknowledge the contribution of the dataset, but I believe that validation experiments on COCO or LVIS are still necessary to demonstrate the effectiveness of the proposed method in this article. After verifying with more datasets, I will consider increasing the score.
5. Visualization can incorporate qualitative validation of some methods to demonstrate comparison with benchmark methods and show the effectiveness of the proposed modules.

[1] Interactive Multi-Class Tiny-Object Detection.

[2] SegGPT: Segmenting Everything In Context.

[3] Semantic-aware SAM for Point-Prompted Instance Segmentation

[4] Beyond Semantic to Instance Segmentation: Weakly-Supervised Instance Segmentation via Semantic Knowledge Transfer and Self-Refinement

**Questions:**

See weakness part.

---

> ### Author Response · Authors · 2024-11-22
> **Response to Reviewer CscQ (Part # 1)**
>
> We sincerely appreciate your detailed and insightful reviews. We hope our response can address your concerns.
>
> > **Q1: a. In Figure 1, "(a)" appears twice b. The method section lacks a paragraph that links together how several modules operate. And the caption in Figure 2 is also very brief. It is not recommended to write the abstract in the form of contribution (1), (2), (3).**
>
> We have revised the attached updated pdf based on your valuable feedback:
> (a) Corrected.
> (b) We have added description of PerSense structure in the caption of Figure 2. In section 1 (L: 104–115) of the paper, we have already provided a para which establishes the connection between the proposed PerSense modules. This content was not replicated in the Methods section to adhere to the page limit and also to avoid any redundancy.
> (c) We have revised the abstract as per your feedback.
>
> > **Q2: In line 43, I did not get the difference between personalized instance segmentation and traditional instance segmentation. If I understand correctly, C3Det(CVPR2022) has a similar idea (even if it is a detection) and needs to be compared and discussed. SegGPT(ICCV2023) also uses some template images to segment the desired objects. Please compare and discuss.**
>
> In PerSense, Class-Label Extractor (CLE) is leveraged to extract the class name from a support or reference image. This extracted class label is provided as a prompt to the grounding detector, enabling it to detect class-specific instances within the query image. By leveraging the class label, the framework first identifies all relevant class instances and selects an initial exemplar based on its similarity with the reference image. This exemplar is then used by the Density Map Generator (DMG) to create a density map, which highlights the spatial distribution of similar instances. The generated density map is processed by the Instance Detection Module (IDM) to identify candidate point prompts, which are subsequently filtered by the Point Prompt Selection Module (PPSM) to retain only those corresponding to instances resembling the support/reference image. In summary, PerSense operates in a funnel-like approach, beginning with personalized detection of the object category using the class label extracted by CLE, followed by exemplar selection for DMG, and finally generating point prompts via IDM and PPSM for the specific visual concept depicted in the reference image.
>
> **Comparison with C3Det [1]:** C3Det is an interactive multi-class tiny object detector and maintains the following key differences with our framework. C3Det is not training-free and requires in-domain training, making it unsuitable for applications where pre-training or domain-specific adjustments are infeasible. Additionally, it is an interactive framework reliant on manual user input, which limits scalability.
> Based on your suggestion, we evaluated the performance of C3Det on the PerSense-D dataset by conducting a cross-dataset generalization test. Specifically, we utilized the C3Det model trained on Tiny-DOTA and assessed its performance on the PerSense-D dataset. The positive location prior in PerSense was used as the initial user input for C3Det to detect similar instances, and the detections were subsequently passed to SAM for segmentation. The performance comparison is summarized in the table below, where PerSense outperformed C3Det by +23.01% mIoU. This result aligns with the performance trends reported by C3Det on the Tiny-DOTA and LCell datasets. As shown in Figure 6 of the C3Det paper, with a single click, the mAP is approximately 63% for Tiny-DOTA and 55% for LCell, calculated at an IoU threshold of 0.5. When transitioning to mIoU, these values naturally decline due to the stricter overlap requirements for segmentation tasks compared to detection tasks.
>
> **Comparison with SegGPT [2]:** Following your suggestion, we also evaluated SegGPT on the PerSense-D dataset using a one-shot approach. PerSense outperformed SegGPT by +16.11% mIoU, demonstrating its superior capability in personalized instance segmentation for dense images. While SegGPT's in-context learning approach excels at general tasks, it struggles to capture the fine-grained contextual relationships needed for the dense scenarios, where densely packed objects of the same class pose a significant challenge. In contrast, PerSense leverages density maps to achieve a more precise spatial understanding of object distribution, enabling improved segmentation performance.
>
> **Performance Comparison Table:**
>
> |   Method   |     Venue      | PerSense-D mIoU |
> |:----------:|:--------------:|:---------------:|
> |    C3Det [1]  |   CVPR 2022|48.60|
> |   SegGPT [2]  |   ICCV 2023| 55.50|
> |  PerSense  |(this work)|71.61|
>
> ---
>
> References:
>
> [1] Interactive multi-class tiny-object detection. CVPR 2022.
>
> [2] Seggpt: Segmenting everything in context. ICCV 2023.

---

> ### Author Response · Authors · 2024-11-22
> **Response to Reviewer CscQ (Part # 2)**
>
> >**Q3: SAPNet uses Point prompt combined with SAM to generate candidate masks, and selects masks that meet specific categories as outputs. This idea is similar to this article, please discuss it. In addition, methods such as Bestie have also predicted peak points as point prompts through affinity maps.**
>
> **Why SAPNet [1] is not comparable?** SAPNet addresses SAM’s inherent semantic ambiguity by incorporating point annotations as prompts to embed semantic information into SAM’s outputs. The framework refines and improves mask proposals to resolve challenges such as distinguishing adjacent objects and avoiding part segmentations. SAPNet requires user-provided point annotations and lacks a mechanism to generalize from a single point annotation to all instances of the same category / visual concept (refer Figure 2 and section 4.2 in SAPNet [1]). Each instance requires its single-point annotation from the user, which contrasts with PerSense’s scope of automatically generating precise, instance-specific point prompts using density maps.
>
> While SAPNet enhances SAM for point-prompted segmentation, it is not directly comparable to PerSense due to its dependency on manual point annotations. Forwarding point prompts generated by PerSense to SAPNet does not constitute a fair comparison, as the automatic generation of instance-specific point prompts from one-shot data is a core contribution of PerSense. However, as an extension of SAM, SAPNet could be integrated into PerSense to further improve SAM’s performance in dense, point-prompted segmentation tasks, leveraging the strengths of both approaches. Presently, PerSense leverages SAM’s basic architecture for its encoder and decoder to ensure fair comparisons with similar approaches like PerSAM. Since PerSense is model-agnostic, frameworks such as SAPNet or any other SAM extensions can still be utilized in place of the basic SAM architecture, providing flexibility for further improvements.
>
> **Why BESTIE [2] is not comparable?**  We believe that the comparison between BESTIE and PerSense is inherently unfair due to their fundamentally different paradigms and design objectives. BESTIE is a training-based, weakly-supervised instance segmentation framework that uses image-level labels as weak supervision. It generates pseudo-instance labels through semantic knowledge transfer, leveraging activation maps from a weakly-supervised semantic segmentation framework. These labels are refined during the training phase using a self-refinement module. Notably, BESTIE requires a maximum of 50,000 training iterations to refine and optimize pseudo-labels for instance segmentation under image-level supervision. In contrast, PerSense is a training-free one-shot segmentation model that leverages pre-trained models, such as SAM to directly perform instance segmentation using one-shot exemplars.
>
> We would also like to emphasize that the recently proposed PerSAM [3], Matcher [4], SegGPT [5] and other prominent one / few-shot segmentation frameworks do not include comparisons with weakly supervised segmentation frameworks, as their underlying design philosophies differ fundamentally.
>
> >**Q4: I acknowledge the contribution of the dataset, but I believe that validation experiments on COCO or LVIS are still necessary to demonstrate the effectiveness of the proposed method. After verifying with more datasets, I will consider increasing the score.**
>
> In addition to PerSense-D, we evaluate our method on COCO-20$^i$ [6] and LVIS-92$^i$ [7], following Matcher [4] data preprocessing and evaluation protocols. Please refer to the table below. We have also included these additional evaluations and comparisons as Appendix C in the updated pdf.
>
> **Comparison with methods involving in-domain training:** To provide a broader perspective, we compare PerSense with both in-domain training methods and training-free approaches. Despite being a training-free framework, PerSense achieves performance comparable to several well-known in-domain training methods as shown in the table below.
>
> **Comparison with PerSAM [3] (point-based prompt method):** PerSense consistently outperforms the recent training-free, point-based PerSAM in segmentation tasks across both sparse datasets (COCO-20$^i$ and LVIS-92$^i$) and the dense dataset (PerSense-D). PerSense demonstrates significant improvements over PerSAM-F, achieving mIoU gains of +25.5% on COCO-20$^i$, +13.4% on LVIS-92$^i$, and +42.2% on PerSense-D.
>
>
> **CONTD..**
>
> References:
>
> [1] Semantic-aware SAM for Point-Prompted Instance Segmentation. CVPR 2024.
>
> [2] Beyond Semantic to Instance Segmentation. CVPR 2022.
>
> [3] Personalize Segment Anything Model with one shot. ICLR 2024.
>
> [4] Matcher: Segment anything with one shot using all-purpose feature matching. ICLR 2024.
>
> [5] Seggpt: Segmenting everything in context. ICCV 2023.
>
> [6] Feature weighting for few-shot segmentation. ICCV 2019
>
> [7] Dataset for large vocabulary instance segmentation. CVPR 2019

---

> ### Author Response · Authors · 2024-11-22
> **Response to Reviewer CscQ (Part # 3)**
>
> > **Q4: Contd..**
>
> **Comparison with Matcher [4] (patch-level and box-based prompt method):**  Matcher achieves superior performance than Persense on sparse datasets like COCO-20$^i$ and LVIS-92$^i$. This increase is due to its reliance on bidirectional patch-level feature matching and bounding box-based prompts which effectively identify distinct object regions in scenarios where objects are sparse and well-separated. In contrast, Matcher struggles with dense images in the PerSense-D dataset due to its reliance on bounding box-based prompts and its relatively limited instance-level matching capabilities, which hinders its performance when segmenting densely packed objects. PerSense outperforms Matcher by +8.8%. This highlights a trade-off between point prompts and bounding box prompts in segmentation performance across sparse and dense images. For sparse images, bounding box prompts are more effective as they encapsulate the entire object, providing more comprehensive information compared to a localized point prompt. However, as discussed in Section 1 (L: 84–89) of the paper, bounding boxes face inherent limitations in dense images due to their fixed shape, inability to effectively address occlusions, and challenges in accommodating object orientation. In such scenarios, point prompts provide superior accuracy, finer control, and greater adaptability, making them more effective in handling occlusions, clutter, and densely packed instances. For this reason, PerSense proposes the automatic generation of precise instance-level point prompts leveraging density maps, rather than relying on bounding box-based prompts.
>
> **Comparison with SegGPT [5] and Painter [6]:**  PerSense outperforms SegGPT on both LVIS-92$^i$ and PerSense-D, achieving a higher mIoU by +7.1% and +16.11%, respectively. However, SegGPT demonstrates superior performance on COCO-20$^i$, likely due to the inclusion of the COCO dataset in its training set. Additionally, PerSense surpasses Painter on COCO-20$^i$ and LVIS-92$^i$ by +15.9% and +15.2% mIoU, respectively, despite Painter having the COCO dataset as part of its training data.
>
> **Additional comments on PerSense (sparse vs dense images):**  PerSense generates point prompts using density maps, which are designed to emphasize the spatial distribution of densely packed objects. On sparse datasets with low object counts, the generated density map often spreads across the entire object. For instance, in the case of a single object, the density map becomes a localized spread concentrated on that object. While this allows PerSense to generate multiple point prompts for the object, it undermines the primary purpose of density maps, which is to capture variations in object density across an image. In such scenarios, density maps provide limited utility, and simpler bounding box-based approaches prove to be more effective. In summary, while PerSense performs reasonably well on sparse datasets like COCO-20$^i$ and LVIS-92$^i$, generating density maps for sparse scenarios (small object count) is less efficient. These cases can be effectively handled by bounding box-based methods, whereas PerSense is specifically designed to excel in dense scenarios by generating precise point prompts where bounding box-based approaches often struggle (already discussed in section 1 of the paper).
>
> ---
>
> | Methods| Venue| COCO-20$^i$ F0| COCO-20$^i$ F1| COCO-20$^i$ F2| COCO-20$^i$ F3| Mean mIoU| LVIS-92$^i$ mIoU| PerSense-D mIoU|
> |---------------|---------------------|----------------|----------------|----------------|----------------|-----------|------------------|-----------------|
> | **In-domain training**|||||||||
> | HSNet [10]|ICCV 21|37.2|44.1|42.4|41.3|41.2|17.4|-|
> | VAT [11]|ECCV 22|39.0|43.8|42.6|39.7|41.3|18.5|-|
> | FPTrans [12]|NeurIPS 22|44.4|48.9|50.6|44.0|47.0|-|-|
> | MIANet [13]|CVPR 23|42.4|52.9|47.7|47.4|47.6|-|-|
> | LLaFS [14]|CVPR 24|47.5|58.8|56.2|53.0|53.9|-|-|
> | **COCO included in training data**|||||||||
> | Painter [8]|CVPR 23|31.2|35.3|33.5|32.4|33.1|10.5|-|
> | SegGPT [5]|ICCV 23|56.3|57.4|58.9|51.7|56.1|18.6|55.5|
> | **Tiny-DOTA as training data**|||||||||
> | C3Det [9]|CVPR 2022|-|-|-|-|-|-|48.6|
> | **Training-free**|||||||||
> | PerSAM [3]|ICLR 24|23.1|23.6|22.0|23.4|23.0|11.5|24.4|
> | PerSAM-F [3]|ICLR 24|22.3|24.0|23.4|24.1|23.5|12.3|29.3|
> | Matcher [4]|-|52.7|53.5|52.6|52.1|52.7|33.0|62.8|
> | PerSense|(this work)|47.8|49.3|48.9|50.1|49.0|25.7|71.6|
>
> ---
>
> References:
>
> [8] A generalist painter for in-context visual learning. CVPR 2023.
>
> [9] Interactive multi-class tiny-object detection. CVPR 2022.
>
> [10] Hypercorrelation squeeze for few-shot segmentation. ICCV 2021.
>
> [11] Cost aggregation with 4d convolutional swin transformer for few-shot segmentation. ECCV 2022
>
> [12] Feature-proxy transformer for few-shot segmentation. NeurIPS 2022.
>
> [13] Aggregating unbiased instance information for few-shot semantic segmentation. CVPR 2023.
>
> [14] When large language models meet few-shot segmentation. CVPR 2024.

---

> ### Author Response · Authors · 2024-11-22
> **Response to Reviewer CscQ (Part # 4)**
>
> >**Q5: Visualization can incorporate qualitative validation of some methods to demonstrate comparison with benchmark methods and show the effectiveness of the proposed modules.**
>
> As suggested, we present step-wise qualitative analysis of PerSense components along with the performance of competing methods in Figure 7, Appendix A.2 of the updated pdf.

---

> > ### Comment · Reviewer_CscQ · 2024-11-29
> >
> > The authors have addressed most of my concerns, so I raise my rate.

---

> > > ### Author Response · Authors · 2024-11-29
> > > **Thanks for your feedback!**
> > >
> > > Thank you for acknowledging our rebuttal and efforts.

---

> ### Author Response · Authors · 2024-11-28
> **Happy to Provide Additional Clarifications**
>
> We sincerely appreciate your thoughtful and thorough review of our paper. We have carefully addressed each of your points and greatly value your constructive feedback, which has contributed to the improvement of our work. As the Author-Reviewer discussion period is nearing its conclusion, please feel free to share any additional concerns or suggestions—we would be more than happy to address them. Thank you.

---

### Official Review · Reviewer_PNZS · 2024-11-03

**Soundness:** 2
**Presentation:** 2
**Contribution:** 3
**Rating:** 6
**Confidence:** 3

**Summary:**

The paper introduces PerSense, a novel, training-free, and model-agnostic framework for personalized instance segmentation in dense images. The contributions of the paper are as follows:

1. New Baseline and Instance Detection Module (IDM): A new baseline is proposed, capable of automatically generating instance-level point prompts, featuring an Instance Detection Module (IDM) that utilizes density maps (DM) to provide candidate point prompts. A density map generator (DMG) highlights the spatial distribution of target objects. Automated Exemplar Selection: A class-label extractor (CLE) and grounding detector are used to automate the selection of effective exemplars, simplifying the DMG’s manual process. Point Prompt Selection Module (PPSM): A Point Prompt Selection Module (PPSM) is designed to reduce false positives within the candidate point prompts. IDM and PPSM are plug-and-play components that integrate seamlessly into the PerSense framework.

2. Feedback Mechanism: A feedback mechanism is introduced to automatically refine exemplar selection based on PerSense’s initial segmentation output, identifying multiple rich exemplars for DMG to improve segmentation accuracy.

3. PerSense-D Dataset: To facilitate personalized segmentation in dense images, the authors introduce PerSense-D, a dataset with 717 densely populated images across 28 object categories, providing a challenging benchmark for future studies.

**Strengths:**

(1) Innovative Training-Free Approach: By using density maps, PerSense avoids the need for extensive training, making it computationally efficient and adaptable to different dense segmentation tasks.
(2) Model-Agnostic Design: The PerSense framework can be seamlessly integrated with various density map generators and grounding detectors, enhancing its flexibility and usability.
(3) Comprehensive Evaluation: Experimental results on the newly introduced PerSense-D dataset show that PerSense significantly outperforms state-of-the-art methods, demonstrating its robustness and high performance in densely packed scenarios.

**Weaknesses:**

1. Methodological Innovation:  Although PerSense's training-free framework demonstrates some level of innovation, its reliance on density maps is not entirely novel, as similar approaches have been applied in traditional vision tasks. While the Instance Detection Module (IDM) and Point Prompt Selection Module (PPSM) bring some originality to the framework, they lack sufficient mathematical derivation or theoretical analysis to substantiate their uniqueness and theoretical advantage over existing methods.
2. Insufficient Baseline Model Comparisons: The experimental comparisons of PerSense are limited to a few general segmentation models (e.g., PerSAM, Matcher) and lack comparisons with other classic indoor segmentation or dense scene methods. For instance, in dense scene segmentation tasks, other clustering-based or feature-matching models might also provide effective solutions. The absence of such baseline model comparisons weakens the demonstration of PerSense's advantage in specific tasks.
3.  Dataset diversity: Although the PerSense-D dataset focuses on dense scenes, it is relatively small in scale (717 images) with limited category coverage, failing to fully represent diverse dense scenes. Real-world applications in personalized segmentation for dense scenes often require larger and more varied samples (e.g., different resolutions, scene types).
4. Experimental Analysis : The ablation studies primarily highlight the incremental effects of IDM, PPSM, and the feedback mechanism, but do not thoroughly explore the synergy among components. For example, the impact of different density map generators (DMG) on model performance is not fully analyzed. Additionally, there is a lack of analysis on the specific contributions of multimodal features (e.g., visual and textual features) in personalized segmentation tasks, making it challenging for readers to understand the role of each feature in the overall performance.

**Questions:**

1. Although the design of PerSense's modules is somewhat innovative, the theoretical support is insufficient. For instance, there is no mathematical proof of how IDM and PPSM improve segmentation accuracy, particularly regarding how they outperform other methods in generating point prompts in dense scenes. Further mathematical derivation would enhance the theoretical foundation of this approach.
2.The experimental validation of this feedback mechanism is limited, as it does not demonstrate the impact of different iteration counts on segmentation accuracy. For example, does varying the initial exemplar selection strategy or iteration count significantly affect model performance? .
3.In dense scenes, challenges like occlusion, lighting changes, and varying object sizes can affect the robustness of segmentation models. However, the paper does not conduct robustness experiments to address these factors. For instance, how does the model perform under low lighting, complex backgrounds, or heavy occlusion?

---

> ### Author Response · Authors · 2024-11-22
> **Response to Reviewer PNZS (Part # 1)**
>
> We sincerely appreciate your detailed and insightful reviews. We hope our response can address your concerns.
>
> **Q1: Although PerSense's training-free framework demonstrates some level of innovation, its reliance on density maps is not entirely novel. Can the originality of IDM and PPSM be further substantiated through mathematical derivations or theoretical analysis?**
>
> PerSense addresses the highly challenging, impactful, and relevant task of personalized segmentation in dense scenarios by automating the generation of instance-level point prompts within a one-shot setting. As discussed in Section 1 (L: 101–104) of the paper, while some initial efforts have explored the use of density maps (DMs) for instance segmentation in natural scenes, a significant gap persists in developing a streamlined approach that explicitly and effectively leverages DMs for automated personalized instance segmentation in dense images. The problem's complexity is underscored by our motivation and extensive experimental results, which clearly demonstrate that even SOTA foundational segmentation models struggle with this task. To the best of our knowledge, PerSense is the only method that leverages density maps while providing an end-to-end, training-free, and model-agnostic one-shot framework for personalized instance segmentation in dense images.
>
> The paper discusses details for IDM and PPSM in Section 3.2 (L: 242-269) and Section 3.3 (L: 270-290), respectively, along with a pseudo-code in Appendix A.1 (L: 727-771). Based on your valuable feedback regarding theoretical analysis, we additionally provide mathematical insights for IDM, starting with composite contour detection via statistical thresholding to the centroid computation for candidate point prompts. We also provide the theoretical rationale behind the formulation of the adaptive threshold in PPSM. We believe that this shall assist in better understanding the design of our proposed modules. **We have also included these insights as Appendix B in the updated PDF.**
>
> ## Instance Detection Module (IDM)
> ---
> ### **Contour Detection and Area Calculation:**
> Contours are identified from the binary image, and the area $ A_{\text{contour}} $ of each contour is calculated. Assuming that the contour areas follow a Gaussian (Normal) distribution, we define:
>
> $$
> A_{\text{contour}} \sim \mathcal{N}(\mu, \sigma^2)
> $$
>
> where:
> - $ \mu $ : Mean area of contours, representing typical object size.
> - $ \sigma $ : Standard deviation, representing variation in contour areas due to size differences among single instances.
>
> The mean $ \mu $ and standard deviation $ \sigma $ are computed as:
>
> $$
> \mu = \frac{1}{N} \sum_{i=1}^N x_i, \quad \sigma = \sqrt{\frac{1}{N} \sum_{i=1}^N (x_i - \mu)^2}
> $$
>
> where $ N $ is the number of detected contours.
>
> ### **Composite Contour Detection Using Statistical Thresholding:**
> To distinguish single-instance contours from composite contours, we set an adaptive threshold based on the Gaussian distribution properties:
>
> $$
> T_{\text{composite}} = \mu + 2\sigma
> $$
>
> This threshold is adopted based on statistical analysis (see Figure 3(b) in the paper) which illustrates that composite contours typically lie beyond $ \mu + 2\sigma $ in the contour area distribution.
>
> $$
> A_{\text{composite}} = \{ A_{\text{contour}} \mid A_{\text{contour}} > T_{\text{composite}} \}
> $$
>
> The probability of a contour being composite is given by:
>
> $$
> P(A_{\text{contour}} > T_{\text{composite}}) = 1 - \Phi\left(\frac{T_{\text{composite}} - \mu}{\sigma}\right)
> $$
>
> where $ \Phi $ is the cumulative distribution function of the standard normal distribution.
>
> ### **Distance Transform for Child Contour Detection:**
> For each composite contour, a distance transform $ D_{\text{transform}} $ is applied to reveal internal sub-regions representing individual object instances:
>
> $$
> D_{\text{transform}}(x, y) = \min_{(i, j) \in K} \| (x, y) - (i, j) \|
> $$
>
> where $ K $ represents contour boundary pixels and  $ x $ and $ y $ are the coordinates of each pixel within the region of interest. A binary threshold applied to $ D_{\text{transform}} $ segments sub-regions within each composite contour, enabling separate identification of overlapping objects in dense scenarios.
>
> ### **Centroid Calculation for Candidate Prompts:**
> For each detected contour (parent and child contours), the centroid is calculated using spatial moments:
>
> $$
> cX = \frac{M_{10}}{M_{00} + \epsilon}, \quad cY = \frac{M_{01}}{M_{00} + \epsilon}
> $$
>
> where $ M_{ij} $ are the spatial moments of the contour, and $ \epsilon $ is a small constant to prevent division by zero. These moments are computed as:
>
> $$
> M_{00} = \sum_{x} \sum_{y} I(x, y), \quad M_{10} = \sum_{x} \sum_{y} x \cdot I(x, y), \quad M_{01} = \sum_{x} \sum_{y} y \cdot I(x, y)
> $$
> where $ I(x, y) $ is the pixel intensity at position $ (x, y) $.
>
> These centroids serve as candidate point prompts, accurately marking the locations of individual object instances in dense scenarios.

---

> ### Author Response · Authors · 2024-11-22
> **Response to Reviewer PNZS (Part # 2)**
>
> > **Q1 contd:**
>
> ## Point Prompt Selection Module (PPSM)
> ---
> The purpose of PPSM’s adaptive threshold is to filter candidate points based on similarity scores, adjusting for object density. This threshold dynamically changes to balance the inclusion of true positives while filtering out false positives in dense scenes. For better understanding, we statistically model the adaptive threshold in PPSM, where the threshold dynamically adjusts according to object count using a fixed scaling factor.
>
> ### **Defining the Adaptive Threshold:**
> Let the cosine similarity scores $ S(x, y) $ (support vs query) at each pixel position $ (x, y) $ form a distribution with the maximum similarity denoted by $ S_{\max} $. For simplicity, we assume that similarity scores across points can be approximated by a Gaussian distribution, with mean $ \mu $ and variance $ \sigma^2 $. The maximum similarity $ S_{\max} $ is then considered the peak or upper bound of this distribution, representing the point with the highest alignment to the target feature. The adaptive threshold $ T $ for point selection is defined as:
>
> $$
> T = \frac{S_{\max} \sqrt{2}}{C}
> $$
>
> where $ C $ represents the object count in the scene. As $ C $ increases, the threshold $ T $ decreases, allowing for a more inclusive selection of points when there is a higher density of objects. We chose $ \sqrt{2} $ as a scaling factor based on empirical results as discussed in Section 5.1.
>
> ### **Probability of Selecting a Point with Similarity Above Threshold:**
> Assuming similarity scores $ S $ follow a Gaussian distribution $ S \sim \mathcal{N}(\mu, \sigma^2) $, the probability $ P $ of a randomly selected point having a similarity score above $ T $ is:
>
> $$
> P(S \geq T) = 1 - \Phi\left(\frac{T - \mu}{\sigma}\right)
> $$
>
> where $ \Phi $ is the cumulative distribution function (CDF) of the standard normal distribution. Substituting for $ T $, we get:
>
> $$
> P(S \geq T) = 1 - \Phi\left(\frac{\frac{S_{\max} \sqrt{2}}{C} - \mu}{\sigma}\right)
> $$
>
> This probability increases as $ C $ grows, implying that a higher object count allows for more points to meet the threshold.
>
> ### **Statistical Balance of True Positives and False Positives:**
> For high values of $ C $, the threshold $ T $ approaches a smaller value. This scaling ensures that PPSM remains inclusive in dense scenes, effectively increasing recall by accepting more points with lower similarity scores. Conversely, for smaller values of $ C $, $ T $ is higher, allowing only points with high similarity scores to pass the threshold. This behavior enhances precision, as fewer points are selected, with a stronger emphasis on high similarity.
>
> By dynamically adjusting $ T $ with $ \frac{S_{\max} \sqrt{2}}{C} $, the adaptive threshold statistically balances true positives and false positives.
>
> ---
> > **Q2: Insufficient baseline model comparisons.**
>
> Following your suggestion, in addition to comparisons with PerSAM, Matcher, and Grounded-SAM, we also evaluate PerSense against C3Det [1], a specialized method for tiny object detection. C3Det is an interactive framework designed to provide bounding boxes for tens or hundreds of tiny objects of a specific class within a given image, based on a single user-provided click on the object of interest. To ensure a fair comparison with our training-free setup, we performed a cross-dataset generalization test, evaluating C3Det's performance on the PerSense-D dataset. Specifically, we utilized the C3Det model trained on Tiny-DOTA [3] and tested it on PerSense-D, leveraging the positive location prior from PerSense as the initial input for C3Det to detect similar instances. The resulting detections were passed to SAM for segmentation.
>
> The results, presented in the table below, demonstrate that PerSense outperformed C3Det by +23.01% mIoU. Additionally, we compared PerSense with SegGPT [2], where PerSense achieved a superior mIoU, outperforming SegGPT on the PerSense-D dataset by +16.11%, further showcasing its robustness in dense segmentation tasks.
>
> Moreover, to ensure a comprehensive evaluation, we compare PerSense with both in-domain training methods and training-free approaches across additional datasets, including COCO-20$^i$ and LVIS-20$^i$. For detailed insights and comparisons, **please refer to the response to Q3.**
>
> | Method   |   Venue        | PerSense-D mIoU |
> |----------|:--------------:|:-----------------:|
> | C3Det [1]    | CVPR 2022    | 48.60           |
> | SegGPT [2]   | ICCV 2023    | 55.50           |
> | PerSense | (this work)  | 71.61           |
>
> References:
>
> [1] Interactive multi-class tiny-object detection. CVPR 2022.
>
> [2] Seggpt: Segmenting everything in context. ICCV 2023.
>
> [3] Object Detection in Aerial Images: A Large-Scale Benchmark and Challenges. TPAMi 2021.

---

> ### Author Response · Authors · 2024-11-22
> **Response to Reviewer PNZS (Part # 3)**
>
> > **Q3 Dataset diversity: Although the PerSense-D dataset focuses on dense scenes, it is relatively small in scale.**
>
> **Why PerSense-D dataset?** As discussed in section 4 of the paper, PerSense-D is a meticulously curated dataset that features significant occlusion, background clutter, and a variety of dense scene types, making it a unique and challenging benchmark for advancing algorithms and developing practical tools for personalized segmentation in dense images. The mean resolution of images is 839 x 967 pixels with a standard deviation of 563 x 610, ranging from a minimum of 250 x 113 pixels to a maximum of 3456 x 4608 pixels, reflecting substantial variation in image resolution. The dataset spans 28 diverse object categories, with an average of 39 object instances per image, totaling 28,395 objects. For a reference, images in LVIS, a popular segmentation dataset contains 3.3 instances per category. This significant difference in object count highlights the exceptional density of the PerSense-D dataset compared to existing benchmarks.
>
> **Evaluation on Additional Datasets:**  To further validate the effectiveness of PerSense, in addition to the PerSense-D dataset, we also evaluated our method on widely used segmentation benchmarks, including COCO-20$^i$ [1] and LVIS-92$^i$ [2]. Please refer to the table below for detailed results. We have also included these additional evaluations and comparisons as Appendix C in the updated pdf.
>
> **Comparison with methods involving in-domain training:**  To provide a broader perspective, we compare PerSense with both in-domain training methods and training-free approaches. Despite being a training-free framework, PerSense achieves performance comparable to several well-known in-domain training methods as shown in the table below.
>
> **Comparison with PerSAM [3] (point-based prompt method):**  PerSense consistently outperforms the recent training-free, point-based PerSAM in segmentation tasks across both sparse datasets (COCO-20$^i$ and LVIS-92$^i$) and the dense dataset (PerSense-D). PerSense demonstrates significant improvements over PerSAM-F, achieving mIoU gains of +25.5% on COCO-20$^i$, +13.4% on LVIS-92$^i$, and +42.2% on PerSense-D.
>
> **Comparison with Matcher [4] (patch-level and box-based prompt method):**  Matcher achieves superior performance than PerSense on sparse datasets like COCO-20$^i$ and LVIS-92$^i$. This increase is due to its reliance on bidirectional patch-level feature matching and bounding box-based prompts which effectively identify distinct object regions in scenarios where objects are sparse and well-separated. In contrast, Matcher struggles with dense images in the PerSense-D dataset due to its reliance on bounding box-based prompts and its relatively limited instance-level matching capabilities, which hinders its performance when segmenting densely packed objects. PerSense outperforms Matcher by +8.8%. This highlights a trade-off between point prompts and bounding box prompts in segmentation performance across sparse and dense images. For sparse images, bounding box prompts are more effective as they encapsulate the entire object, providing more comprehensive information compared to a localized point prompt. However, as discussed in Section 1 (L: 84–89) of the paper, bounding boxes face inherent limitations in dense images due to their fixed shape, inability to effectively address occlusions, and challenges in accommodating object orientation. In such scenarios, point prompts provide superior accuracy, finer control, and greater adaptability, making them more effective in handling occlusions, clutter, and densely packed instances. For this reason, PerSense proposes the automatic generation of precise instance-level point prompts leveraging density maps, rather than relying on bounding box-based prompts.
>
> **Comparison with SegGPT [5] and Painter [6]:**
> PerSense outperforms SegGPT on both LVIS-92$^i$ and PerSense-D, achieving a higher mIoU by +7.1% and +16.11%, respectively. However, SegGPT demonstrates superior performance on COCO-20$^i$, likely due to the inclusion of the COCO dataset in its training set. Additionally, PerSense surpasses Painter on COCO-20$^i$ by +15.9% and +15.2% mIoU, despite Painter having the COCO dataset as part of its training data.
>
> References:
>
> [1] Feature weighting and boosting for few-shot segmentation. ICCV 2019.
>
> [2] Dataset for large vocabulary instance segmentation. CVPR 2019.
>
> [3] Personalize Segment Anything Model with one shot. ICLR 2024.
>
> [4] Matcher: Segment anything with one shot using all-purpose feature matching. ICLR 2024.
>
> [5] Seggpt: Segmenting everything in context. ICCV 2023.
>
> [6] Images speak in images: A generalist painter for in-context visual learning. CVPR 2023.

---

> ### Author Response · Authors · 2024-11-22
> **Response to Reviewer PNZS (Part # 4)**
>
> > **Q3 Contd..**
>
> | Methods| Venue| COCO-20$^i$ F0| COCO-20$^i$ F1| COCO-20$^i$ F2| COCO-20$^i$ F3|     Mean mIoU     | LVIS-92$^i$ mIoU| PerSense-D mIoU|
> |---------------|-------------------------------|----------------|----------------|----------------|----------------|-----------|------------------|-----------------|
> | ***In-domain training***|||||||||
> | HSNet [8]|ICCV 21|37.2|44.1|42.4|41.3|41.2|17.4|-|
> | VAT [9]|ECCV 22|39.0|43.8|42.6|39.7|41.3|18.5|-|
> | FPTrans [10]|NeurIPS 22|44.4|48.9|50.6|44.0|47.0|-|-|
> | MIANet [11]|CVPR 23|42.4|52.9|47.7|47.4|47.6|-|-|
> | LLaFS [12]|CVPR 24|47.5|58.8|56.2|53.0|53.9|-|-|
> | ***COCO included in training data***|||||||||
> | Painter [6]|CVPR 23|31.2|35.3|33.5|32.4|33.1|10.5|-|
> | SegGPT [5]|ICCV 23|56.3|57.4|58.9|51.7|56.1|18.6|55.5|
> | ***Tiny-DOTA as training data***|||||||||
> | C3Det [7]|CVPR 2022|-|-|-|-|-|-|48.6|
> | ***Training-free***|||||||||
> | PerSAM [3]|ICLR 24|23.1|23.6|22.0|23.4|23.0|11.5|24.4|
> | PerSAM-F [3]|ICLR 24|22.3|24.0|23.4|24.1|23.5|12.3|29.3|
> | Matcher [4]|-|52.7|53.5|52.6|52.1|52.7|33.0|62.8|
> | PerSense|(this work)|47.8|49.3|48.9|50.1|49.0|25.7|71.6|
>
> ---
>
> > **Q4: The ablation studies primarily highlight the incremental effects of IDM, PPSM, and the feedback mechanism, but do not thoroughly explore the synergy among components.**
>
> To further emphasize the interaction and synergy among the PerSense components, we already provided a step-by-step qualitative analysis in Appendix A.2 of the paper. We have improved this qualitative analysis in the updated manuscript by introducing comparison with SOTA methods. Please see Appendix A.2 of the updated pdf. This analysis demonstrates how each component works in coordination with others, complementing their functionalities to collectively enhance the overall segmentation performance in dense scenarios. Additionally, to showcase the model-agnostic capability of the PerSense framework, we have already presented results with two different density map generators (DMG), DSALVANet [13] and CounTR [14], in Table 1 of the paper. Furthermore, the performance of each PerSense component is analyzed separately for DSALVANet and CounTR in Table 2(a).
>
> >**Q5: Understanding the role of multimodal features.**
>
> In PerSense, multimodal features are utilized specifically during class-label extraction (CLE), where textual prompts are derived from visual inputs in the support set. These textual prompts guide the grounding detector by providing class labels to detect the specific object category in the query image. Besides this step, the entire PerSense framework leverages only visual features for tasks such as density map generation, instance detection, and point prompt refinement. While multimodal features play a limited role in PerSense, their effective integration during CLE ensures accurate and personalized segmentation guidance. In future work, we plan to extend the use of multimodal features to other components of PerSense to further improve segmentation performance.
>
> > **Q6: Although the design of PerSense's modules is somewhat innovative, the theoretical support is insufficient.**
>
> Please see response to Q1.
>
> ---
>
> References:
>
> [7] Interactive multi-class tiny-object detection. CVPR 2022.
>
> [8] Hypercorrelation squeeze for few-shot segmentation. ICCV 2021.
>
> [9] Cost aggregation with 4d convolutional swin transformer for few-shot segmentation. ECCV 2022
>
> [10] Feature-proxy transformer for few-shot segmentation. NeurIPS 2022.
>
> [11] Aggregating unbiased instance information for few-shot semantic segmentation. CVPR 2023.
>
> [12] When large language models meet few-shot segmentation. CVPR 2024
>
> [13] Few-shot object counting with dynamic similarity-aware in latent space. IEEE TGRS 2024.
>
> [14] Countr: Transformer-based generalised visual counting. BMVC 2022.

---

> ### Author Response · Authors · 2024-11-22
> **Response to Reviewer PNZS (Part # 5)**
>
> > **Q7: The experimental validation of this feedback mechanism is limited, as it does not demonstrate the impact of different iteration counts on segmentation accuracy.**
>
> **Multiple Iterations in Feedback Mechanism:** The feedback mechanism in PerSense utilizes the initial segmentation output from the decoder to select multiple exemplars for refining the density map via DMG. This process occurs in a single pass, with exemplars selected based on their SAM scores, and does not involve multiple iterations, effectively fixing the iteration count at one. An ablation study, presented below, examines the effect of multiple iterations in feedback mechanism on segmentation accuracy as well as computational efficiency. The results indicate that additional iterations are unnecessary, as they do not improve segmentation accuracy beyond the results achieved in the single pass but instead increase computational overhead, reducing the efficiency of the PerSense pipeline. Intuitively, this is because the first-pass exemplars (four in our case) correspond to the most confident instances of the target object category. These exemplars are easily detected by DMG, with their boundaries well delineated by SAM, even when using a single initially selected exemplar as input. In subsequent iterations, the same exemplars are repeatedly selected due to their distinct visual features and consistently high SAM scores, attributed to the clearly defined boundaries in their segmentation masks. Consequently, multiple feedback iterations provide no additional benefit, rendering further iterations redundant. We have also included this ablation in Appendix D of the updated pdf.
>
> | No. of iterations (Feedback Mechanism) | PerSense mIoU | Average Inference time per image (sec) |
> |:--------------------------------------:|:-------------:|:-------------------------------------:|
> | 1                                      |     71.61     |                 2.7                  |
> | 2                                      |     71.65     |                 3.1                  |
> | 3                                      |     71.63     |                 3.5                  |
> | 4                                      |     71.60     |                 3.9                  |
>
>
>
> > **Q8: In dense scenes, challenges like occlusion, lighting changes, and varying object sizes can affect the robustness of segmentation models. how does the model perform under low lighting, complex backgrounds, or heavy occlusion?**
>
> We recognize the importance of evaluating challenges such as occlusion, lighting variations, and diverse object sizes in dense scenes. As discussed in section 4 of the paper, the PerSense-D dataset was specifically curated with these challenges in mind, focusing on heavy occlusion, background clutter, and varying object sizes, making it a unique and challenging benchmark for dense image segmentation. These are precisely the scenarios that the PerSense-D dataset is designed to address. Furthermore, to establish the robustness of our model, we have also provided results on large-scale datasets such as COCO-20$^i$ and LVIS-92$^i$, as discussed in response to Q3. These datasets feature low lighting conditions and complex backgrounds, ensuring that our model is thoroughly evaluated for its performance under challenging real-world scenarios.

---

> ### Comment · Reviewer_PNZS · 2024-11-25
> **Response to the authors' rebuttal**
>
> Although this foundational model-based research paradigm seems to have gradually gained popularity, the authors provided detailed and convincing explanations to address my concerns and made certain contributions.

---

> > ### Author Response · Authors · 2024-11-25
> > **Thanks for your feedback!**
> >
> > Thank you for acknowledging our rebuttal and efforts.

---

### Author Response · Authors · 2024-11-22
**General Response and Summary of Updates to Manuscript**

We thank the reviewers **(PNZS, CscQ, GosF, L45P)** for their positive and thoughtful feedback, and we appreciate the comments aimed at improving our work. We summarize the main points presented in our response and hope that we have satisfactorily addressed all the concerns raised by reviewers.

- We provide additional mathematical insights for the proposed instance detection module (IDM) starting with composite contour detection via statistical thresholding to the centroid computation for candidate point prompts. We also provide a detailed theoretical rationale behind the formulation of the adaptive threshold in the point prompt selection module (PPSM) (see Appendix B in the updated pdf).

- We include additional comparisons of our PerSense with C3Det [1], Painter [2], SegGPT [3] in addition to Matcher [4], PerSAM [5], and Grounded-SAM [6] (see Appendix C in the updated pdf).
- In addition to the PerSense-D dataset, we also evaluate our PerSense on standard segmentation datasets, specifically COCO-20$^i$ [7] and LVIS-92$^i$ [8], and conduct comparisons with a diverse range of approaches, including both in-domain training methods and training-free techniques (see Appendix C in the updated pdf).

- We conduct additional ablation studies, including the impact of multiple iterations in the proposed feedback mechanism, a component-wise analysis of PerSense on the COCO dataset, and an evaluation of PerSense's memory consumption (see Appendix D in the updated pdf).

- We enhanced Figure 7 to provide a detailed step-by-step component-wise qualitative analysis of PerSense, incorporating qualitative validation of benchmark methods to demonstrate the effectiveness of the proposed modules (see Appendix A.2 in the updated pdf).

- We address the questions raised by reviewers and provide clarifications on the feedback resulting in the improvement of our work.

Please do not hesitate to reach out if you need any further clarifications. Thank you.

---

References:

[1] Interactive multi-class tiny-object detection. CVPR 2022.

[2] Images speak in images: A generalist painter for in-context visual learning. CVPR 2023.

[3] Seggpt: Segmenting everything in context. ICCV 2023.

[4] Matcher: Segment anything with one shot using all-purpose feature matching. ICLR 2024.

[5] Personalize Segment Anything Model with one shot. ICLR 2024.

[6] Grounded sam: Assembling open-world models for diverse visual tasks. arXiv 2024.

[7] Feature weighting and boosting for few-shot segmentation. ICCV 2019.

[8] Dataset for large vocabulary instance segmentation. CVPR 2019.

---

### Meta-Review · Area_Chair_6YLA · 2024-12-21

**Metareview:**

This submission proposed a new method and dataset for personalized instance segmentation in dense images. The approach performs well using a model-agnostic framework, and the overall effectiveness is verified. The authors also highlight their contribution as the first to explicitly propose the concept of dense image segmentation. The weaknesses of the paper include the novelty issue of using the density map for segmentation, missing some experimental analysis, and the small scale and unclear evaluation scheme of the PerSense-D dataset. After rebuttal, the authors addressed some concerns, but the concerns about the dataset still exist. A small-scale dataset without an evaluation scheme design could result in overfitting results by heavy manual adjustment of the models or pipelines. Considering these concerns, the overall recommendation is reject.

**Additional Comments On Reviewer Discussion:**

This submission received four reviews.
Reviewer PNZS's major concern was mainly about the dense map's novelty issue, comparison with classical dense segmentation approaches, and some missing experimental analysis. The authors basically addressed the reviewer's concerns, and the score was improved from 5 to 6.
Reviewer CscQ suggested comparison with more works, including C3Det, SegGPT, SAPNet, and Bestie, and more experiments on COCO and LVIS. The author addressed these concerns and the score is improved from 5 to 6.
Reviewer GosF's concern is mainly about the dataset and the benchmark, which lacks a proper evaluation scheme and has the risk of overfitting due to the small scale and no subset division. After rebuttal, the authors addressed some other concerns but the dataset issue still remains. The review score is improved from 3 to 5.
Reviewer L45P's major concern was a 4-shot misunderstanding, which was clarified, and the score (6) remained unchanged after the rebuttal.
Overall, three reviewers suggest that this paper is marginally above the threshold, and one reviewer suggests it is marginally below the threshold. Considering the dataset issue is still unsolved, the final recommendation is reject.

---

### Decision · Program_Chairs · 2025-01-22

Reject